# A pair of congenic mice for imaging of transplants by positron emission tomography using anti-transferrin receptor nanobodies

Thomas Balligand[1,2]*, Claire Carpenet[1,3], Sergi Olivé Palau[1,4], Tom Jaspers[5], Pavana Suresh[1], Xin Liu[1], Himadri Medhi[1], Yoon Ho Lee[1], Mohammad Rashidian[6], Bart De Strooper[7], Hidde L Ploegh[1]*, Maarten Dewilde[5,7]*

[1]Program for Cellular and Molecular Medicine, Boston Children's Hospital, Harvard Medical School, Boston, United States; [2]Unité de Recherche en Physiologie Moléculaire, Namur Research Institute for Life Sciences (NARILIS), Université de Namur, Namur, Belgium; [3]CBS2 University of Montpellier, Montpellier, France; [4]University of Barcelona, Barcelona, Spain; [5]Laboratory for Therapeutic and Diagnostic Antibodies, KU Leuven Campus Gasthuisberg O and N2, Leuven, Belgium; [6]Dana-Farber Cancer Institute, Harvard Medical School, Boston, United States; [7]VIB Center for Brain and Disease Research, KU Leuven Campus Gasthuisberg O and N5, Leuven, Belgium

**\*For correspondence:**
thomas.balligand@unamur.be (TB);
hidde.ploegh@childrens.harvard.edu (HLP);
maarten.dewilde@kuleuven.be (MD)

## eLife Assessment

In this highly innovative study, Carpenet C et al explore the use of nanobody-based PET imaging to track proliferative cells after in vivo transplantation in mice, in a fully immunocompetent setting. The development of a unique set of PET tracers and mouse strains to track genetically-unmodified transplanted cells in vivo is an **important** novel asset that could potentially facilitate cell tracking in different research fields. The evidence provided is **compelling** as the new method proposed might facilitate overcoming certain limitations of alternative approaches, such as full sized immunoglobulins and small molecules.

**Abstract** Two anti-transferrin receptor (TfR) nanobodies, $V_HH123$ specific for mouse TfR and $V_HH188$ specific for human TfR, were used to track transplants non-invasively by PET/CT in mouse models, without the need for genetic modification of the transferred cells. We provide a comparison of the specificity and kinetics of the PET signals acquired when using nanobodies radiolabeled with $^{89}Zr$, $^{64}Cu$, and $^{18}F$, and find that the chelation of the $^{89}Zr$ and $^{64}Cu$ radioisotopes to anti-TfR nanobodies results in radioisotope release upon endocytosis of the radiolabeled nanobodies. We used a knock-in mouse that expresses a TfR with a human ectodomain (Tfrc$^{hu/hu}$) as a source of bone marrow for transplants into C57BL/6 recipients and show that $V_HH188$ detects such transplants by PET/CT. Conversely, C57BL/6 bone marrow and B16.F10 melanoma cell line transplanted into Tfrc$^{hu/hu}$ recipients can be imaged with $V_HH123$. In C57BL/6 mice impregnated by Tfrc$^{hu/hu}$ males, we saw an intense $V_HH188$ signal in the placenta, showing that TfR-specific $V_HHs$ accumulate at the placental barrier but do not enter the fetal tissue. We were unable to observe accumulation of the anti-TfR radiotracers in the central nervous system (CNS) by PET/CT but showed evidence of CNS accumulation by radiospectrometry. The model presented here can be used to track many transplanted cell types by PET/CT, provided cells express TfR, as is typically the case for proliferating cells such as tumor lines.

## Introduction

Non-invasive tracking of specific cell types in vivo is a desirable goal in many fields of biomedical research, particularly as it pertains to immunology and cancer. Knowledge of the biodistribution of immune cells and/or tumor cells in a disease model is essential to monitor therapeutic interventions. Real-time non-invasive in vivo imaging may be achieved by several methods, each with their strengths and limitations. Fluorescence and luminescence-based methods suffer from absorbance and dispersal of emitted light (*Yoon et al., 2022*), which limits the depth at which images of acceptable resolution can be obtained. Fluorescence-based methods in conjunction with multi-photon microscopy provide cellular resolution, but usually require invasive surgery to gain access to the cells of interest (*Choi et al., 2015*). Methods that rely on the use of radioisotopes, such as single photon emission computed tomography (SPECT) and positron emission tomography (PET) do not suffer those drawbacks. While SPECT and PET lack the resolution of optical microscopy, they have the advantage of being non-invasive, quantitative, and enabling whole-body imaging in small animals at a resolution of ~1 mm, or a 1 microliter volume, for PET (*Serkova et al., 2021*) and even lower than 1 mm for SPECT (*van der Have et al., 2009*). These methods are finding increasing use in the field of tumor immunology, as they can provide a whole-body image, unlike other approaches.

Current PET/CT cell tracing methods require the use of a cell-specific tracer. This typically relies on the detection of an endogenous surface marker that is specific for the cells of interest (*Cao et al., 2019*), or by genetically engineering cells of interest to enable their visualization, for example through expression of a viral kinase of unique specificity (*Gambhir et al., 2000*), or by introduction of a surface marker that can be selectively visualized (*Minn et al., 2019*).

$V_H$Hs, also termed nanobodies, are enjoying increasing use for the generation of immuno-PET tracers that yield images of a quality superior to what is achieved using regular, intact immunoglobulins (*Harmand and Islam, 2021*). Nanobodies are the recombinantly expressed variable fragments ($V_H$) of heavy chain-only immunoglobulins produced by camelids (*Hamers-Casterman et al., 1993*; *Ingram et al., 2018*). Their small size (~15 kDa), superior tissue penetration, specificity, affinity, and much shorter circulatory half-life compared to intact immunoglobulins make nanobodies excellent tracers for in vivo imaging. We here apply nanobodies specific for the transferrin receptor to track the fate of transplanted cells non-invasively.

The transferrin receptor (TfR; CD71; encoded by *Tfrc*) is a homodimeric type-II transmembrane protein that is near-ubiquitously expressed, in particular on proliferating cells (*Kawabata, 2019*). This includes many tumor cells with some variability depending on the tumor type and differentiation status (*Ryschich et al., 2004*; *Faulk et al., 1980*; *Daniels et al., 2012*; *Basuli et al., 2017*; *Rychtar-cikova et al., 2017*; *Kindrat et al., 2016*; *Horniblow et al., 2017*; *Shirakihara et al., 2022*; *Prutki et al., 2006*). The TfR binds to and endocytoses iron ($Fe^{+++}$)-loaded transferrin (Tf). Tf remains bound to the TfR in endosomal compartments, where the resident low pH releases the $Fe^{+++}$ cargo and thus converts Tf into apoTf, which remains TfR-bound at endosomal pH. From there, TfR-bound apoTf returns to the cell surface, where the TfR releases apoTf at neutral pH. TfR is also expressed by endothelial cells that line the blood-brain barrier (BBB) for delivery of $Fe^{+++}$-loaded transferrin to the central nervous system by transcytosis. The nanobodies used in this study bind TfR and can traverse the BBB (*Wouters et al., 2020*; *Wouters et al., 2022*). Mouse embryos critically depend on an iron supply in the form of Tf captured from the maternal circulation, delivered to the embryo via the TfR expressed on the syncytiotrophoblast-I (*Sangkhae and Nemeth, 2019*). Whether the small size of nanobodies allows them by analogy with the BBB- to traverse the placenta and reach the embryo has not been explored.

We use two nanobodies: $V_H$H123 (also termed Nb62 *Wouters et al., 2020*) and $V_H$H188, which recognize the murine TfR and the human TfR (TFRC), respectively. We use these $V_H$Hs together with a knock-in mouse model that expresses a TfR with a human TfR ectodomain (Tfrc*hu/hu*) (*Wouters et al., 2022*). The specificity of each nanobody for the respective TfR and the availability of Tfrc*hu/hu* mice as a source of primary cells allows us to track different cell types in a transplant setting. $V_H$H123 allows the detection of cells of mouse origin, such as bone marrow progenitors or tumor cells, transplanted into Tfrc*hu/hu* mice. Conversely, we use $V_H$H188 to track primary cells from Tfrc*hu/hu* mice after transfer into wild-type mice. Pregnancy represents a unique model akin to a transplant. We report the first immuno-PET study on localization of the TfR in mouse embryos in live mice.

We find that $^{89}$Zr and $^{64}$Cu, ligated to these TfR-specific V$_H$Hs by non-covalent chelation, are released from the imaging agent upon binding the TfR, due to internalization and exposure to low endosomal pH. The release of free $^{89}$Zr and $^{64}$Cu from imaging agents are factors to consider when using these over extended periods of time. Covalent modification of V$_H$Hs with $^{18}$F avoids the release of free radio-isotope, but the short half-life of $^{18}$F limits the observation window to <12 hr. We compare the use of three commonly used positron-emitting isotopes ($^{89}$Zr, $^{64}$Cu, and $^{18}$F) and offer a suite of tools to track many cell types in mice, without the requirement for specific cell markers or genome editing. At most, crossing of a mouse model of interest with the Tfrc$^{hu/hu}$ mouse line would be required (deposited at Jackson Laboratories, strain 038212).

## Results

### V$_H$H123 binds mouse TfR and V$_H$H188 binds human(ized) TfR in vitro and in vivo

We characterized the specificity of V$_H$H123 (anti-mouse TfR) and V$_H$H188 (anti-human TfR) by biochemical methods and by PET/CT. We confirmed the specificity of each anti-TfR V$_H$H for its target by immunoprecipitation from lysates of HEK293 (human) or B16.F10 (mouse) cell lines. We show that both V$_H$Hs bind only to the appropriate TfR, with no obvious cross-reactivity to other surface-expressed proteins by immunoblot, LC/MSMS analysis of immunoprecipitates, SDS-PAGE of $^{35}$S-labeled proteins and flow cytometry (*Figure 1*; *Table 1*; *Figure 1—figure supplement 1*). Virtually all contaminants that were co-immunoprecipitated were of cytoplasmic and nuclear localization (*Table 1*; *Supplementary file 1*). This is of minor concern, as these proteins are not accessible to our anti-TfR nanobodies in an in vivo setting.

For PET experiments, we attached a deferoxamine (DFO)-azide moiety to each V$_H$H via a sortase A-catalyzed transpeptidation reaction (*Rashidian et al., 2017*). The V$_H$H-DFO-azide adducts were then modified with dibenzocyclooctyne-polyethyleneglycol$_{20kDa}$ (DBCO-PEG$_{20kDa}$) using click chemistry (*Figure 2*; *Figure 2—figure supplements 1–4*). The PEG$_{20kDa}$ moiety, hereafter referred to as 'PEG,' serves to extend the half-life of the V$_H$H in the circulation and decreases non-specific kidney uptake (*Rashidian et al., 2017*). The PEG-DFO-modified V$_H$Hs were labeled with $^{89}$Zr through chelation by DFO (see methods) to generate the V$_H$H-PEG-DFO-$^{89}$Zr radiotracers. We injected the V$_H$H-PEG-DFO-$^{89}$Zr conjugates into C57BL/6 mice and Tfrc$^{hu/hu}$ mice and collected PET/CT images at various times after retro-orbital injection. Injection of the V$_H$H123-based conjugate into Tfrc$^{hu/hu}$ mice yielded a signal exclusively in the kidneys, which we consider a non-specific signal (*Figure 3C*). $^{89}$Zr radiolabeled V$_H$Hs typically accumulate in the kidneys in the absence of a specific target. Similarly, injection of the V$_H$H188-based conjugate into C57BL/6 mice likewise yielded a strong PET signal only in the kidneys (*Figure 3B*), again considered non-specific accumulation of the tracer.

For the properly matched V$_H$H-mouse combinations ($^{89}$Zr radiolabeled V$_H$H123 conjugate injected into C57BL/6 mice; $^{89}$Zr radiolabeled V$_H$H188-conjugate injected into Tfrc$^{hu/hu}$ mice), each produced an intense PET signal that localizes to bones, mainly the vertebrae, sacrum, coxal bone as well as both epiphyses of the femur and the proximal epiphysis of the humerus at 48 hr post-radiotracer injection (*Figure 3A and D*). The mean activity measured by PET was significantly different between matched and unmatched V$_H$H/mouse pairs in the liver, spleen, femoro-tibial articulation (knee), and spleen, by region of interest (ROI) analysis (*Figure 3E–F*). While the signals from long bone extremities and vertebrae suggest uptake of the anti-TfR radioconjugate by bone marrow (*Zimmer et al., 2011*), presumably due to the high demand for iron required for erythropoiesis (*Richard and Verdier, 2020*), this pattern of label distribution is better explained by sequestration of free $^{89}$Zr in mineralized bone and cartilage (*Raavé et al., 2019*; *Abou et al., 2011*) (see also *Figure 3—figure supplement 2*). The weak signal from the diaphyses of the femora relative to the intense signal in the vertebral column and bone extremities suggests that accumulation of free $^{89}$Zr in mineralized bone predominates under these conditions. To assess this, we measured the activity of specific tissues ex vivo 72 hr post radiotracer injection (*Figure 3G–H*). Several tissues showed significant differences in activity between the matched and unmatched mouse/V$_H$H pairs: particularly in the brain, spleen, small intestine, and liver. Femur bones flushed of all bone marrow retained a very strong activity in the matched conditions – showing that the mineral bone retains a high activity when depleted of bone marrow (>200,000 CPM/bone) – confirming the deposit of free $^{89}$Zr in the bone matrix. Nevertheless, flushed bone marrow

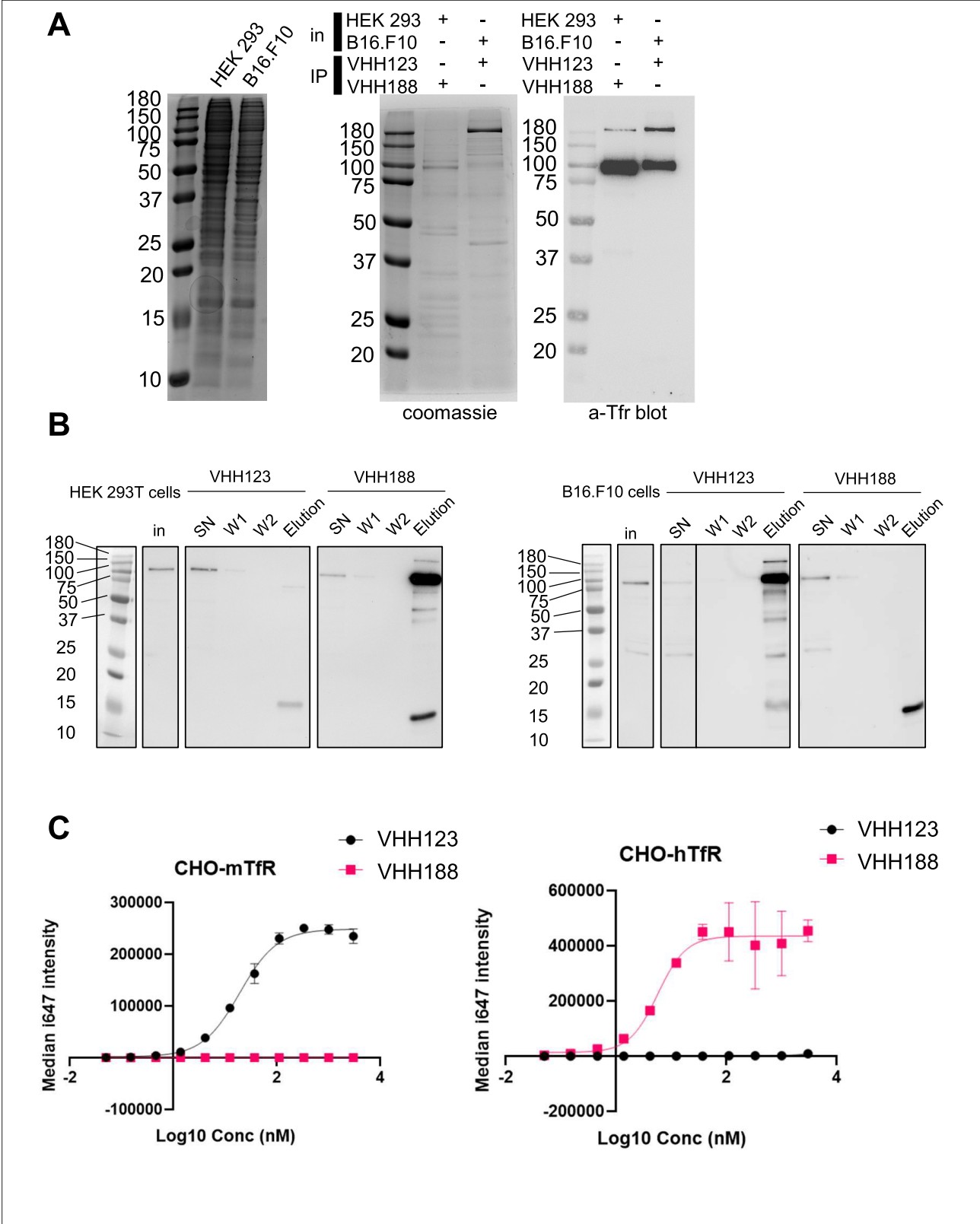

**Figure 1.** Binding specificities of VHH123 and VHH188. (**A**) Left-most panel: Coomassie staining of an SDS-PAGE gel showing the input cell lysates used in immunoprecipitation experiments depicted in the next panels. Middle panel: lysates as shown on left-side panel were incubated with nanobody-coated paramagnetic beads overnight. Beads were then washed five times then boiled in SDS sample buffer before loading on SDS-PAGE. This panel shows Coomassie staining of SDS-PAGE of the bead eluates. In: input lysates. IP: Nanobody used for immunoprecipitation. Right panel: the

*Figure 1 continued on next page*

*Figure 1 continued*

same eluates as in middle panel were run on SDS-PAGE gel then transferred to a PVDF membrane. Membrane was blocked, stained using a mouse monoclonal anti-transferrin receptor (TfR) (cross-reactive for human and mouse), washed then stained with an anti-mouse-HRP secondary monoclonal. See Methods section for more details. Experiment repeated a total of two times. (**B**) Cell lines were incubated with $^{35}$S-labeled Met before performing the same immunoprecipitation procedure as described in A, with the exception that the beads were washed only twice before elution in Laemmli buffer. SDS-PAGE was run with input cell lysate (in), unbound fraction (SN), washes (W1 and W2), and eluates for each condition before transfer to a PVDF membrane. Membrane was blocked, stained using a mouse monoclonal anti-TfR (cross-reactive for human and mouse), washed then stained with an anti-mouse-HRP secondary monoclonal. Experiment repeated a total of two times. (**C**) Flow cytometry characterization of the specificity of $V_HH123$ and $V_HH188$. CHO cells overexpressing either the mouse isoform of TfR (mTfr, left panel) or human isoform (hTfr, right panel) were labeled with serial dilutions of either Flag-tagged $V_HH123$ or $V_HH188$. I647-fluorescently labeled anti-FLAG IgG was used to detect the presence of either $V_HH$ at the cell surface by flow cytometry. Experiment repeated a total of two times.

The online version of this article includes the following source data and figure supplement(s) for figure 1:

**Source data 1.** PDF file containing original gels and western blots for *Figure 1A and B*, indicating the relevant bands and treatments.

**Source data 2.** Original files for gels and western blots displayed in *Figure 1A and B*.

**Figure supplement 1.** Autoradiograph of $^{35}$S labeled lysates of HEK 293T cells and B16.

**Figure supplement 1—source data 1.** PDF file containing the original autoradiograph film scan of *Figure 1—figure supplement 1*, indicating the relevant bands and treatments.

**Figure supplement 1—source data 2.** Original file of the autoradiograph film scan of *Figure 1—figure supplement 1*.

showed activity (20,000–30,000 CPM/flushed bone) that was significantly higher in the matched vs. unmatched conditions (*Figure 3G*). We conclude that at later timepoints, the strong PET signals in the long bone extremities (femur, tibia, and humerus) are due in part to bone marrow accumulation of anti-TfR radiolabeled $V_HHs$, but predominantly so to accumulation of free $^{89}$Zr in the mineralized

**Table 1.** Summary of peptides identified from LC/MS/MS analysis of the gel sections from *Figure 1A*, middle panel. Transferrin receptor 1 is highlighted in bold. See *Supplementary file 1* for full dataset.

| | Species | Unique | Total | Protein | UniProt | Gene | mw (kDa) | Coverage % | Subcell. Loc. |
|---|---|---|---|---|---|---|---|---|---|
| | | **87** | **273** | **Transferrin receptor protein 1** | **P02786** | **TFRC** | **84.82** | **68.60%** | **Membrane** |
| | | 54 | 189 | Nucleolin | P19338 | NCL | 76.57 | 46.80% | Cytoplasm |
| | | 51 | 76 | Nucleolar RNA helicase 2 | Q9NR30 | DDX21 | 87.29 | 56.70% | Nucleus |
| | | 44 | 50 | Epiplakin | P58107 | EPPK1 | 555.32 | 11.90% | Cell junction |
| | | 42 | 57 | ATP-dependent RNA helicase A | Q08211 | DHX9 | 140.87 | 36% | Mitochondrion |
| | | 42 | 50 | ATP-dependent RNA helicase DHX30 | Q7L2E3 | DHX30 | 133.85 | 39.50% | Nucleus |
| | | 36 | 45 | Myosin-9 | P35579 | MYH9 | 226.39 | 21.60% | Cytoplasm |
| VHH188/HEK cell lysate | human | 31 | 63 | 60 S ribosomal protein L4 | P36578 | RPL4 | 47.67 | 51.50% | Cytoplasm |
| | | 237 | 623 | Myosin-9 | Q8VDD5 | Myh9 | 226.23 | 69.20% | Cytoplasm |
| | | 128 | 244 | CAD protein | B2RQC6 | Cad | 243.08 | 59.60% | Nucleus |
| | | 79 | 111 | Unconventional myosin-Va | Q99104 | Myo5a | 215.4 | 44.60% | n/a |
| | | 73 | 96 | Dedicator of cytokinesis protein 7 | Q8R1A4 | Dock7 | 241.29 | 39.70% | Unknown |
| | | 60 | 68 | Plectin | Q9QXS1 | Plec | 533.86 | 18.10% | Cell junction |
| | | 55 | 257 | Actin, cytoplasmic 1 | P60710 | Actb | 41.71 | 66.90% | Cytoplasm |
| | | **55** | **156** | **Transferrin receptor protein 1** | **Q62351** | **Tfrc** | **85.68** | **59.80%** | **Membrane** |
| VHH123 /B16 .F10 cell lysate | mouse | 50 | 58 | Dystonin | Q91ZU6 | Dst | 833.7 | 9.60% | Cytoplasm |

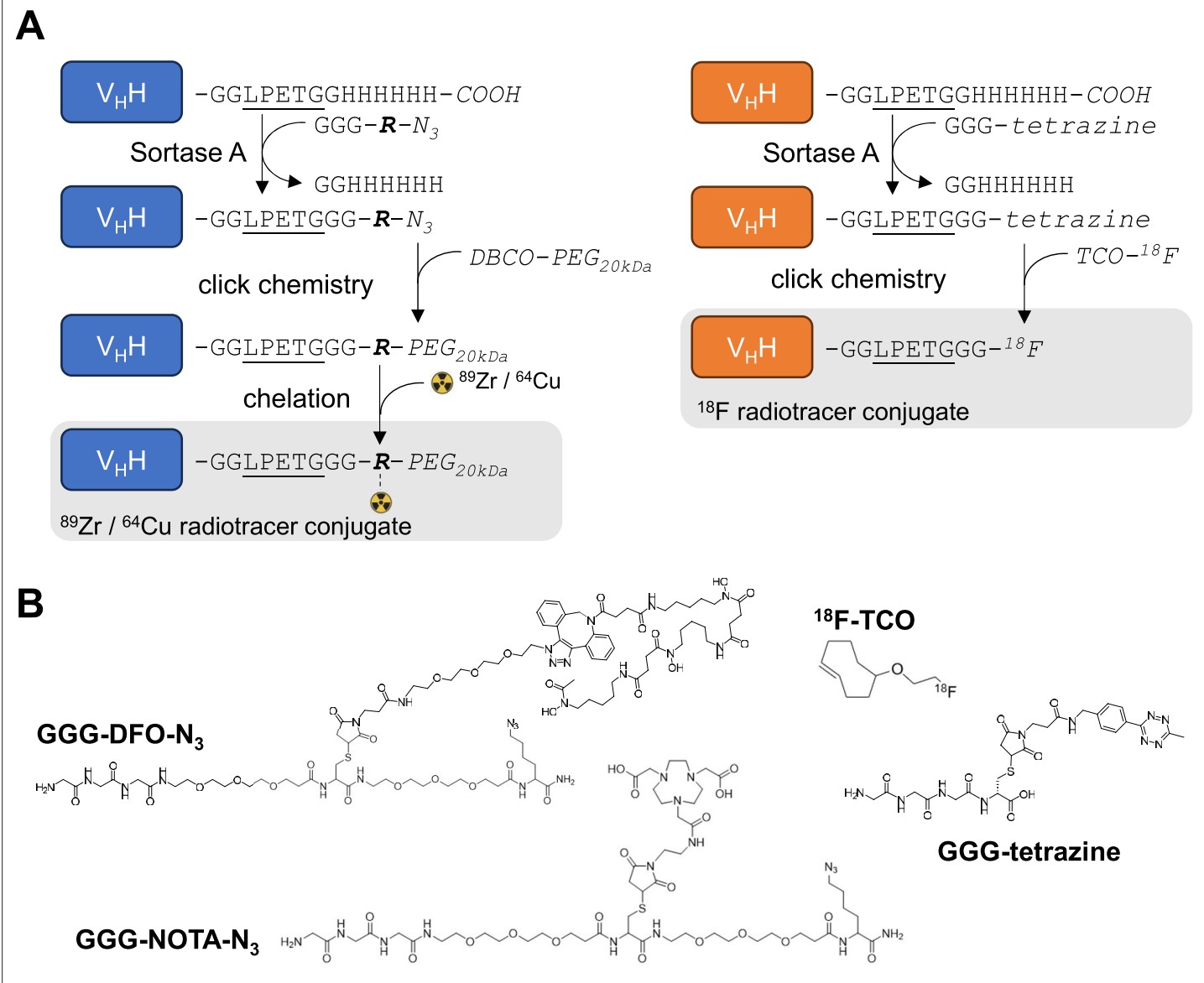

**Figure 2.** Radiolabeling strategy. (**A**) Schematic of sortase A and click-chemistry steps to generate each radio-conjugate used in this study. Capital letters denote amino acids, unless if written in *italic* where they denote chemical groups or elements. Bold italic 'R' represents either DFO (deferoxamine) or NOTA (2,2′,2″-(1,4,7-triazacyclononane-1,4,7-triyl)triacetic acid). DBCO: dibenzocyclooctyne, PEG$_{20kDa}$: polyethylene-glycol (20kDa mw), TCO: trans-cyclooctene. Underlined is the LPETG Sortase A cleavage site consensus motif. See *Figure 2—figure supplement 3* for detailed methods. (**B**) Structures of GGG-nucleophiles used in sortase A-mediated conjugations (GGG-DFO-N$_3$, GGG-NOTA-N$_3$, and GGG-tetrazine). The $^{18}$F-TCO click-chemistry partner of GGG-tetrazine is also depicted. These structures were synthesized as described methods and *Figure 2—figure supplements 5 and 6*.

The online version of this article includes the following source data and figure supplement(s) for figure 2:

**Figure supplement 1.** Schematic representation of each step required to generate $^{89}$Zr-based (blue), $^{64}$Cu-based (green), and $^{18}$F-based (orange) V$_H$H conjugates.

**Figure supplement 2.** SDS-PAGE followed by Coomassie Blue stain, showing the individual constructs used throughout this work, as purified and concentrated post-sortase A transpeptidation and post DBCO-PEG$_{20kDa}$ click-chemistry conjugation when performed.

**Figure supplement 2—source data 1.** PDF file containing the original gel of *Figure 2—figure supplement 2*, indicating the relevant bands and treatments.

**Figure supplement 2—source data 2.** Original file for the gel presented in *Figure 2—figure supplement 2*.

**Figure supplement 2—source data 3.** LC/MS mass measurement data for compounds shown in *Figure 2—figure supplement 2*.

**Figure supplement 3.** Synthetic schemes of labeling reagents.

*Figure 2 continued on next page*

*Figure 2 continued*
**Figure supplement 4.** Mass spectra of the labeling reagents shown *Figure 2—figure supplement 3*.
**Figure supplement 5.** Radio-Thin Layer Chromatography QC data for VHH-$^{18}$F constructs.
**Figure supplement 6.** Semi-preparative HPLC gamma chromatogram of 18F-TCO (collected from 8.5 to 9.2 min).

bone matrix. No obvious accumulation of tracer was seen in brain parenchyma by PET, but significant differences of activity were found in the ex vivo analysis (*Figure 3H*), in line with the reported ability of V$_H$Hs that recognize the TfR to deliver bound materials across the blood-brain barrier (*Wouters et al., 2020*). The failure to detect a PET signal of adequate strength in the brain are due to the comparatively small amounts of anti-TfR V$_H$H that cross the BBB. When analyzing PET images obtained with V$_H$Hs that recognize targets other than the TfR, which includes V$_H$Hs that recognize CD8 (*Rashidian et al., 2017*), Ly6C/G (*Pishesha et al., 2022*), or CD11b (*Priem et al., 2020*), as examined previously, no accumulation of $^{89}$Zr in skeletal elements was seen. These previous observations, combined with the lack of any bone signal in the mismatched anti-TfR V$_H$H/mouse pairs, lead us to conclude that free $^{89}$Zr is released from the V$_H$H-conjugates in vivo, but only when the V$_H$H binds to its respective TfR.

Because chelation of $^{89}$Zr to DFO depends on its charge (*Imura et al., 2021*), we hypothesized that binding of V$_H$H-$^{89}$Zr conjugates to the appropriate TfR delivers them to endosomal compartments of low pH, where $^{89}$Zr is then released from the DFO moiety. We examined release of $^{89}$Zr from the V$_H$H123-PEG-DFO construct in vitro by exposure to low pH (*Figure 3—figure supplement 1*). $^{89}$Zr is freed from the adduct at pH <5.6, which is well within the pH range of late endosomal compartments (*Cain et al., 1989*). Whether the mildly acidic pH of early or recycling endosomal compartments is sufficient for isotope release in vivo remains to be determined.

## Comparison of $^{89}$Zr, $^{64}$Cu, and $^{18}$F isotopes conjugated to the anti-TfR V$_H$Hs

Having observed the release of $^{89}$Zr from the V$_H$H123-PEG-DFO imaging agent, we considered the use of $^{64}$Cu as an alternative radioisotope. We also synthesized an adduct covalently labeled with $^{18}$F to circumvent any possible release of free radioisotope. The rather different half-lives of $^{89}$Zr (~3.3 days), $^{64}$Cu (~12 hr), and $^{18}$F (~110 min) dictate the observation windows allowed by their use, which we arbitrarily set at 3–5 half-lives, thus ranging from 10 hr ($^{18}$F) to ~2 weeks ($^{89}$Zr), with ~3% of the injected dose remaining after 5 half-lives due to isotope decay.

We generated a V$_H$H123-PEG-NOTA-$^{64}$Cu version, where $^{64}$Cu is chelated by NOTA and would not accumulate in mineralized bone if released. The use of a $^{64}$Cu-labeled tracer should allow an observation window of 3–5 half-lives, i.e., ~36–60 hr. For comparison, we produced a V$_H$H123-tetrazine-TCO-$^{18}$F construct, where $^{18}$F is incorporated covalently through tetrazine-TCO click-chemistry, by an inverse electron-demand Diels–Alder (IEDDA) reaction between V$_H$H123-tetrazine and $^{18}$F-TCO (*Selvaraj and Fox, 2013*; *Figure 2—figure supplements 5 and 6*). This precludes isotope release other than by proteolysis. We injected these imaging agents for a side-by-side comparison with V$_H$H123-PEG-DFO-$^{89}$Zr in C57BL/6 mice (*Figures 4 and 5*). At 1 hr post-injection of the $^{89}$Zr, $^{64}$Cu, or $^{18}$F-labeled radiotracer, we observe a strong PET signal in the diaphyses and both epiphyses of the femora and in the coxal bones and sacrum (bone marrow), thus showing that V$_H$H123 accumulates specifically at those locations, and that the signal is not simply due to free radioisotope only (*Figure 4*). $^{89}$Zr and $^{64}$Cu radiolabeled V$_H$H123 also accumulated in the spleen. The $^{18}$F-based imaging agent provided a sharper picture, with some signal originating from the gut and gall bladder, typically seen when using $^{18}$F-based radiotracers comprising bulky hydrophobic moieties, in this case a TCO-Tetrazine clicked product (*Zhou et al., 2020*). At the 12 hr timepoint the $^{18}$F signal had fully decayed due to the short half-life of $^{18}$F. Improved signal quality is thus offset by the shorter half-life of $^{18}$F. For the $^{64}$Cu-based agent, the bone marrow signal remains weakly visible at 24 hr post-injection (*Figure 5*). The liver and gut signals increase at later timepoints for the $^{64}$Cu-labeled tracer, a phenomenon attributable to a slow release of $^{64}$Cu from NOTA to plasma proteins such as albumin, which delivers copper to the liver where it can then be conjugated to ceruloplasmin (*Dearling et al., 2011*; *Mirick et al., 1999*). In mice that received the $^{89}$Zr-labeled agent, we mostly observe a signal from the bone extremities and vertebrae at 24 hr post-injection, which we attribute to labeling of mineralized bone with free $^{89}$Zr with a minor contribution of a TfR-specific bone marrow signal. For all imaging agents, we observe a pair

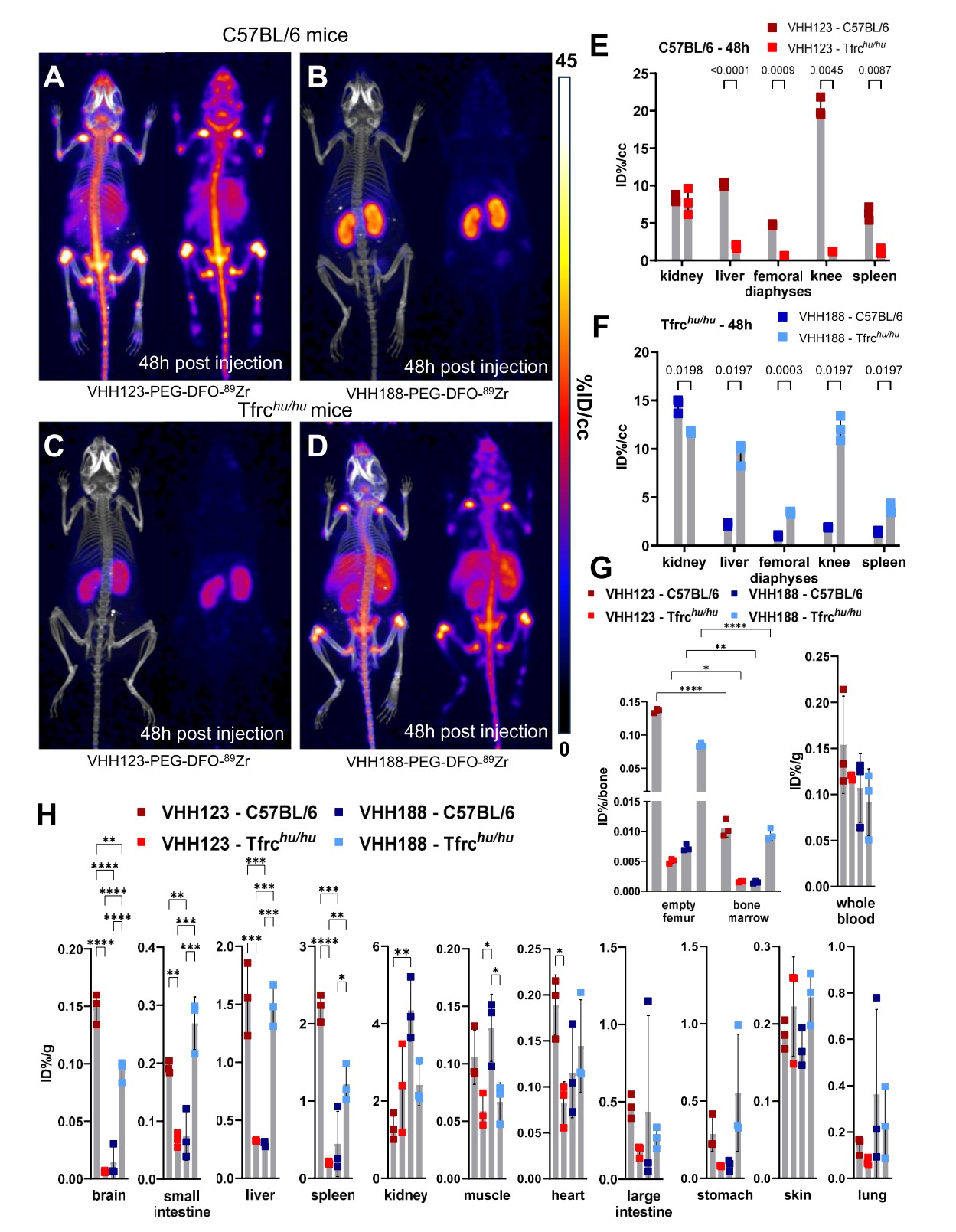

**Figure 3.** Biodistribution of [89]Zr-labelled anti-TfR VHHs. (**A–D**) C57BL/6 and Tfrc[hu/hu] mice were injected with 3.7 MBq (100 µCi) of either V$_H$H123-PEG(20 kDa)-DFO-[89]Zr or V$_H$H188-PEG(20 kDa)-DFO-[89]Zr by retro-orbital injection. The mice were imaged by PET/CT at several timepoints post-injection. Shown here at the maximum intensity projection images acquired at 48 hr post-injection of the conjugate. Each panel comprises maximum intensity projection (MIP) overlayed with CT signal on the left, and PET MIP alone on the right. PET intensity scale is displayed on the right (%ID/cc).

*Figure 3 continued on next page*

*Figure 3 continued*

All repeats (=1 mouse) shown are from one experiment out of two replicates. The VHH123 /Tfrc*hu* and VHH188/C57BL6 groups were used as negative controls for VHH binding. N=3 mice per group. (**A**) C57BL/6 mouse injected with the V$_H$H123-based conjugate. (**B**) C57BL/6 mouse injected with the V$_H$H188-based conjugate. (**C**) Tfrc*hu/hu* mouse injected with the V$_H$H123-based conjugate. (**D**) Tfrc*hu/hu* mouse injected with the V$_H$H188-based conjugate. Experiment performed with 3 mice per condition, with one mouse shown as representative of each condition. (**E**) Region Of Interest (ROI) analysis of images acquired from mice as shown in A and C and all repeats thereof. The mean ID%/cc is plotted for each ROI and mouse repeat. (**F**) Same as E, but for images acquired from mice as shown in B and D and all repeats thereof. (**G**) Left graph: Ex vivo activity measurement of flushed femurs (thus mineral bone) and the bone marrow they contained, 72 hr post radiotracer injection as in A-D. Each dot represents measurement of one mouse on a scale of injected dose percentage per bone (ID%/bone). Right graph: ex vivo activity measurement of 20 µL of whole blood. Each dot represents the activity from one mouse, on a scale of injected dose percentage per gram of tissue (ID%/g). Bars show SD. (**H**) Ex vivo activity measurements from different tissues, performed at 72 hr post-radiotracer injection. No capillary depletion was performed. Each dot represents one measurement from one mouse on a scale of ID%/g. Bars show SD.

The online version of this article includes the following source data and figure supplement(s) for figure 3:

**Source data 1.** PET/CT images of all repeats (mice) from the experiment shown in *Figure 3*.

**Figure supplement 1.** 185 MBq (5 mCi) of $^{89}$Zr-oxalate stock solution was obtained from the Cyclotron Lab at UW Madison, USA.

**Figure supplement 2.** PET/CT of one female C57BL/6 mouse that received 2.775 MBq (75 µCi) of $^{89}$Zr as a free element (unchelated/unconjugated) alongside 75×10$^8$ gold chiral nano-particles.

of punctiform signals in the anterior region of the cranium. These appear very prominently with the $^{18}$F tracer, and at later timepoints -but less clearly- with the $^{89}$Zr and $^{64}$Cu imaging agents, potentially explained by the lower positron emission range of $^{18}$F. Limited by the resolution of the CT images, we tentatively attribute these signals to the accumulation of the radiotracer in the roots of the incisors, where iron uptake is required for proper amelogenesis (*McKee et al., 1987*).

We conclude that internalization of $^{89}$Zr and $^{64}$Cu V$_H$H123-based imaging agents leads to release of free radioisotope ($^{89}$Zr and $^{64}$Cu), which is responsible for most of the signal detected at later (24 hr or later) timepoints and thus does not truly reflect distribution of the TfR itself at those timepoints. This is particularly clear from a ROI analysis of the liver and knee PET signals, when normalized to the average kidney signal (*Figure 5B*), that shows a shift of distribution of PET signal towards the liver with $^{64}$Cu-labeled V$_H$H123 and towards the knee for $^{89}$Zr-labeled VHHs at later timepoints post-injection. Combined, these results establish specificity of recognition of the respective V$_H$Hs for their intended targets. Moreover, the results show that release of $^{89}$Zr or $^{64}$Cu from the imaging agents is the unavoidable consequence of their binding to the proper targets. The choice of isotopes for conjugation to anti-TfR nanobodies must, therefore, be made with respect to the tissue(s) that are to be visualized, as well as the time scale of the experiment (~3–5 isotope half-lives).

## $^{89}$Zr-conjugated V$_H$H123 tracks transplanted bone marrow cells in vivo

Having established that V$_H$H123 accumulates in the bone marrow in vivo, we tested whether V$_H$H123 could detect bone marrow cells transplanted into lethally irradiated recipient mice. To enable the specific detection of the donor cells, we isolated bone marrow cells from C57BL/6 mice and transplanted 1.5×10$^6$ cells into lethally irradiated (10 Gy) Tfrc*hu/hu* recipient mice by retro-orbital injection. We used V$_H$H123-PEG-DFO-$^{89}$Zr to determine whether bone marrow engraftment would suffice to reproduce the signal pattern observed in C57BL/6 wild-type mice. At 2 weeks post bone marrow transplantation, 24 hr after injection of V$_H$H123-PEG-DFO-$^{89}$Zr, the presence of the transferred cells in the marrow is observed in the spleen and femoral bone, together with a strong signal from free $^{89}$Zr accumulation in the bone matrix (*Figure 6A*). The signal in the spleen appeared more prominent than that observed when imaging wild-type C57BL/6 mice at 24 hr post-injection of the same VHH, as pointed out through ROI analysis (*Figure 6F* vs. *Figure 5B*), which could reveal the presence of hematopoietic bone marrow in the spleen. The same observations were made in the reverse experiment where transplantation of Tfrc*hu/hu* bone marrow into C57BL/6 recipients was performed before imaging the recipients 15 days after transplantation using V$_H$H188-PEG-DFO-$^{89}$Zr (*Figure 6B and G*). Strikingly, the images acquired at 1 hr post V$_H$H188-PEG-DFO-$^{89}$Zr injection show a powerful signal in the spleen, something not observed when imaging Tfrc*hu/hu* mice with the same radiotracer, which can be easily interpreted as the presence of bone marrow engraftment in the spleen (*Figure 6C and G*). As a control, V$_H$H188-PEG-DFO-$^{89}$Zr was unable to show engraftment of C57BL/6 bone marrow cells in an isogenic recipient (*Figure 6D and G*). Because the transferred bone marrow cells proliferated in

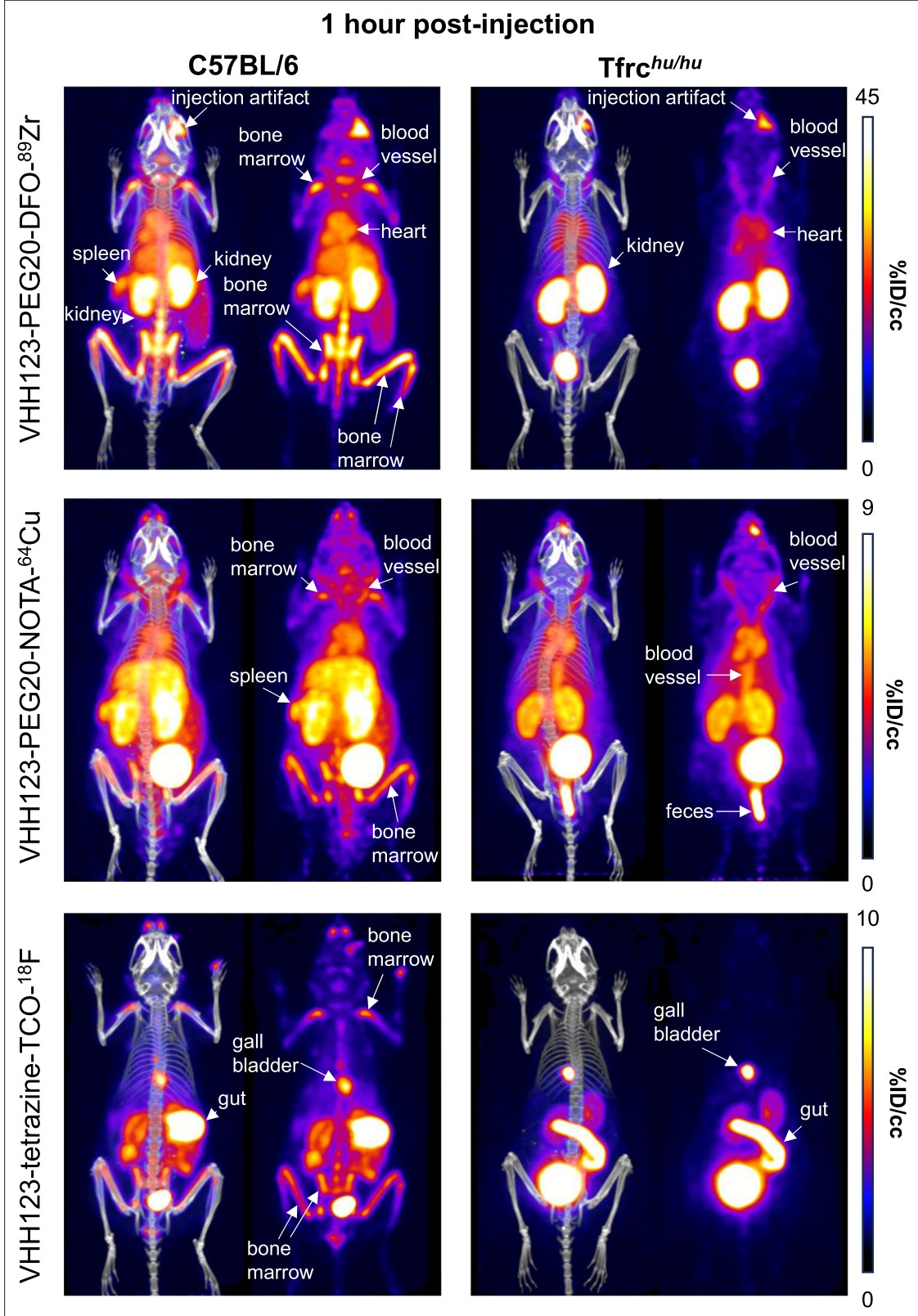

**Figure 4.** Comparison of the PET signals acquired 1 hour after injection of $^{89}$Zr, $^{64}$Cu and $^{18}$F-labelled VHH123. C57BL/6 mice (left column) and Tfrc$^{hu/hu}$ mice (right column) were injected with 3.7 MBq (100 µCi) of different V$_H$H-123-PEG(20 kDa) based conjugates as indicated on the left side of the panels. PET/CT images were acquired for each condition at 1 hr post-injection. Each panel comprises maximum intensity projection (MIP) overlayed with CT signal on the left, and PET MIP alone on the right. Positron emission tomography (PET) intensity scales are displayed on the right of each row

*Figure 4 continued on next page*

*Figure 4 continued*

(%ID/cc). This figure pools the representative pictures obtained from three independently performed sets of experiments where one specific $V_H$H123-radiolabeled conjugate was tested per experiment. Three C57BL/6 mice were imaged in each experiment for each condition, and two Tfrc$^{hu/hu}$ mice were imaged in each experiment for each condition, save for the $^{64}$Cu condition where three Tfrc$^{hu/hu}$ mice were imaged. Experiments using the $^{89}$Zr and $^{64}$Cu radioisotopes have been replicated twice, and the experiment using $^{18}$F was performed once. All groups with Tfrc$^{hu/hu}$ mice were used as negative controls for $V_H$H123 binding.

The online version of this article includes the following source data for figure 4:

**Source data 1.** PET/CT images of all repeats (mice) from the experiment shown in **Figure 4**.

the 14 days prior to imaging, they reached numbers adequate for the release of $^{89}$Zr that would then accumulate in the bone matrix. Indeed, imaging of Tfrc$^{hu/hu}$ recipient mice immediately after C57BL/6 bone marrow transplantation using $V_H$H188-PEG-DFO-$^{89}$Zr reveals only a faint signal in the knee and liver (**Figure 6E**). Specific tracing of TfR-positive cells of mouse or human origin is thus possible when using $V_H$H123 or $V_H$H188 as the tracer in a Tfrc$^{hu/hu}$ or wild-type recipient, respectively. Depending on the mass of TfR$^+$ cells present the recipients, timing of tracer injection and the PET imaging session is critical to avoid the confounding effect of the free $^{89}$Zr bone matrix signal. These data establish the feasibility of detecting transplanted bone marrow cells and their hematopoietic descendants amidst populations or recipient cells that remain invisible to the imaging agent used.

## TfR-positive B16.F10 melanoma cells are detected by $^{64}$Cu-conjugated $V_H$H123

Next, we asked whether we could trace tumor cells that express mouse TfR, but that are not derived from bone marrow hematopoietic cells. $5 \times 10^4$ B16.F10 mouse melanoma cells were injected *i.v.* by tail vein injection to generate metastatic lung tumors in Tfrc$^{hu/hu}$ mice. We allowed the B16.F10 cells to engraft and establish metastases for up to 4 weeks post-transplantation. We injected the recipients with $V_H$H123-PEG-NOTA-$^{64}$Cu at weeks 2 and 4. Imaging done at 2 weeks post-inoculation of the B16F10 tumor did not yield a clear signal in the lungs (**Figure 7A**), as metastases typically arise some 3 weeks after injection for this number of cells. At 4 weeks post-transplantation, we observed a clear signal originating from the lungs of the same mice (**Figure 7C**). No signal was detected in the no tumor control group (**Figure 7E**). To exclude the possibility of non-TfR-specific accumulation of radiotracer in necrotic tumor tissue, we injected tumor-bearing mice with a non-specific anti-GFP $V_H$HEnh-PEG-NOTA-$^{64}$Cu conjugate (**Figure 7B and D**), which showed only weak passive accumulation in the lung metastases at 4 weeks post-tumor cell infusion. At necropsy, we confirmed the presence of melanotic tumors in the lungs (**Figure 7F**), which were also visible on the lung CT. One mouse also had a small tumor in the liver, and another in the skin of the left flank. No mice had tumors in the heart or kidneys. By ROI analysis of the PET signals, $V_H$H123 gave a significantly higher signal in the whole lungs (**Figure 7G**). The PET signal from B16.F10 metastases in the lung correlates well with what appears to be dense tumor tissue on the corresponding CT image. When performing ROI analysis restricted to the PET signal in hyperdense lung tumor tissue, $V_H$H123-PEG-NOTA-$^{64}$Cu gave an overall stronger signal when compared to $V_H$HEnh-PEG-NOTA-$^{64}$Cu, although not by a significant margin (**Figure 7H**). It is important to emphasize that ROI analysis on lung and heart is less precise, due to breathing movements and beating heart during acquisition of the PET signal. Kidney signals were significantly different between tumor-free mice that received $V_H$H123-PEG-NOTA-$^{64}$Cu and the other groups, which we attribute to a difference in clearance rate of the radioconjugates. The anti-mouse TfR nanobody $V_H$H123 is thus well-suited to track TfR-expressing cells of mouse origin in Tfrc$^{hu/hu}$ recipient mice. Since all proliferating cells express TfR, in principle, any tumor or proliferating cell of mouse origin can be detected non-invasively without the need for genetic modification of the transplanted cells. This congenic pair of mice, in combination with the species specificity of the anti-TfR $V_H$Hs, is thus a unique tool for studies of this type.

## $^{89}$Zr-conjugated VHH188 binds to TfR at the blood-placenta barrier as observed by PET/CT

Pregnancy presents a unique situation akin to a transplant setting. Maternal Tf is taken up at the blood-placenta barrier (BPB) to deliver iron to the developing embryo. We asked whether the $V_H$H188

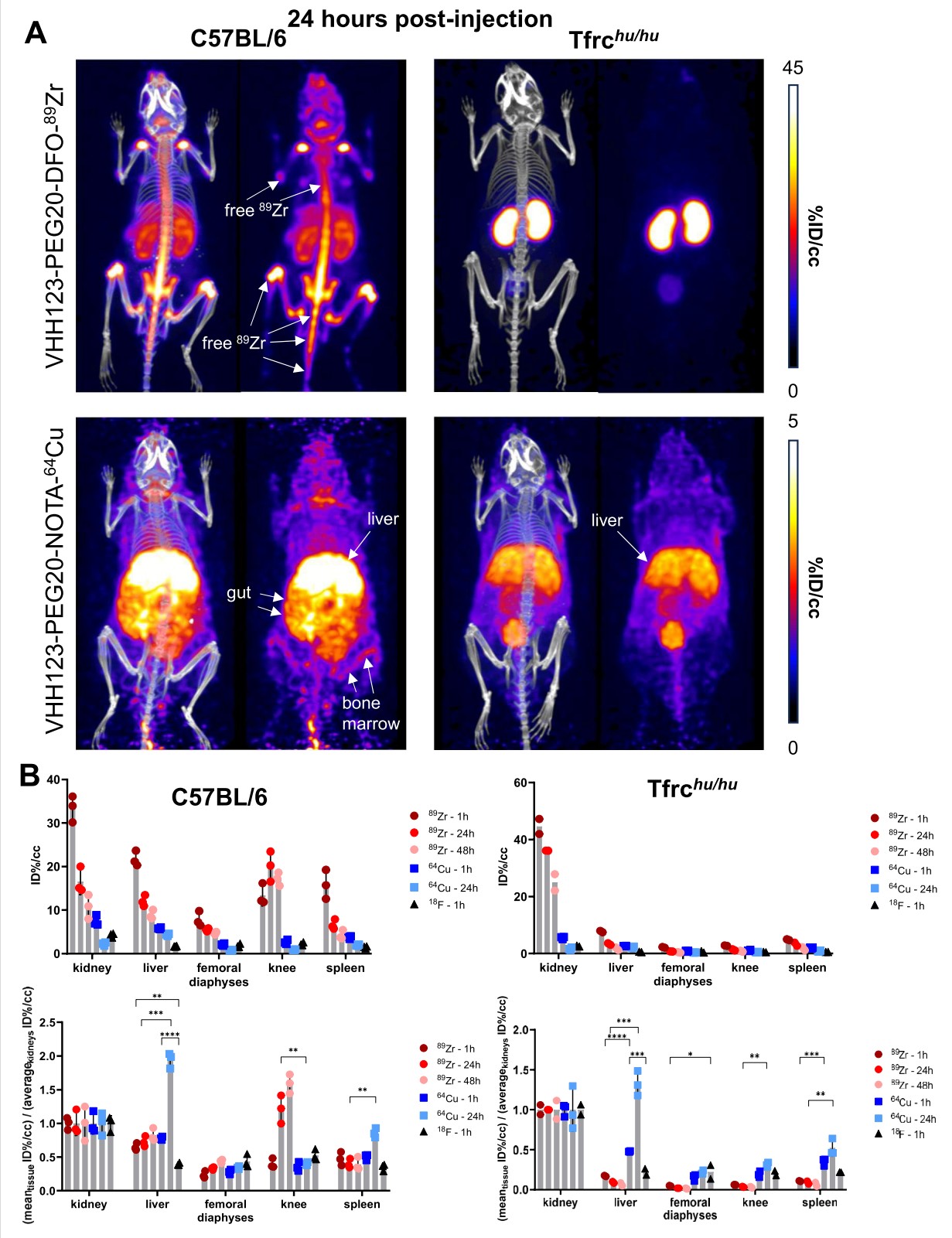

**Figure 5.** Comparison of the PET signals acquired 24 hours after injection of [89]Zr and [64]Cu-labelled VHH123. (**A**) PET/CT images of the same mice as shown in *Figure 4* but acquired at 24 hr post-injection. Each panel comprises maximum intensity projection (MIP) overlaid with CT signal on the left, and PET MIP alone on the right. Positron emission tomography (PET) intensity scales are displayed on the right of each row (%ID/cc). This figure pools the representative pictures obtained from three independently performed sets of experiments where one specific $V_H$H123-radiolabeled conjugate was

*Figure 5 continued on next page*

*Figure 5 continued*
tested per experiment. Three C57BL/6 mice were imaged in each experiment for each condition, and two Tfrc*hu/hu* mice were imaged in each experiment for each condition, save for the $^{64}$Cu condition where three Tfrc*hu/hu* mice were imaged. Experiments using the $^{89}$Zr and $^{64}$Cu radioisotopes have been replicated twice, and the experiment using $^{18}$F was performed once. All groups with Tfrc*hu/hu* mice were used as negative controls for V$_H$H123 binding.
(**B**) Region of interest (ROI) analysis of images acquired from mice as shown in all panels of *Figures 4 and 5A*, and all repeats thereof, and organized as: left column – C57BL/6, right column – Tfrc*hu/hu*. Top row: each point represents the mean ID%/cc for one mouse. Bottom row: same data as in the graph above, but each point represents the mean ID%/cc of a specific tissue normalized to the average ID%/cc values found in the kidneys of the same group.

The online version of this article includes the following source data for figure 5:

**Source data 1.** PET/CT images of all repeats (mice) from the experiment shown in *Figure 5*.

nanobody would detect TfR expressed by syncytiotrophoblasts-I at the BPB. C57BL/6 females were mated with a Tfrc*hu/hu* male or with a C57BL/6 male as a control. Embryos will thus be heterozygous for the presence of the two TfR isoforms. We hypothesize that proper mouse and human TfR homodimers will be formed, in addition to the possible formation of the interspecific hybrid TfR heterodimer. Two weeks post-fertilization, the pregnant females were injected retro-orbitally with V$_H$H188-PEG-DFO-$^{89}$Zr. We observed rapid uptake of the V$_H$H188-PEG-DFO-$^{89}$Zr radiotracer in the individual placentas of the heterozygous (Tfrc*hu/wt*) embryos (*Figure 8A*). Wild-type C57BL/6 placentas were barely detectable in females crossed with a C57BL/6 male, showing a much weaker signal (*Figure 8B*). This difference is significant through ROI analysis (*Figure 8C*). We attribute the low but detectable signal for V$_H$H188 to possible cross-reactivity to the mouse TfR when expressed at the very high levels on syncytiotrophoblast-I: placental tissue expresses a much higher than average level of Tfr (*Sangkhae and Nemeth, 2019*; *Cox et al., 2009*). Post-euthanasia dissection and separation of the placenta from the embryo was done, followed by a separate round of PET/CT imaging of the extirpated uterus and the embryos it contained. This confirmed that the intense V$_H$H188-PEG-DFO-$^{89}$Zr signal originates from the placenta and not from the embryo (*Figure 8D and E*). V$_H$H188-PEG-DFO-$^{89}$Zr thus detects Tfrc*hu/wt* placental tissue. Notwithstanding intense labeling of the placenta, we did not see a signal that corresponds to free $^{89}$Zr in either embryos or in the pregnant female. This results, therefore, departs from what was seen for any of the other imaging experiments, all of which showed release of $^{89}$Zr when the appropriate target TfR was expressed. Of note, the signal in the placenta was already quite intense at 1 hr post-injection of the radiotracer (*Figure 8F*).

## Discussion

The ability to track the fate of transplanted cells non-invasively over time is a useful asset for several applications. These include the monitoring of tumor growth in response to various forms of therapy in a pre-clinical setting, or to follow the fate of transplanted cells of hematopoietic origin. Methods currently in use include luminescence-based approaches, in which the transplanted cells are engineered to express a suitable reporter, such as a luciferase (*Yoon et al., 2022*; *de Almeida et al., 2011*). Luminescence methods do not provide cellular resolution. Alternatively, to achieve single-cell resolution, fluorescence-based methods have been applied in multi-photon microscopy (*Choi et al., 2015*), but this typically involves invasive surgical interventions to expose the target tissue or organ, because absorption by surrounding tissue limits the depth of penetration of the excitation beam and emitted fluorescence.

Non-invasive methods such as NMR lack the specificity to detect particular cell populations, although tumors that exceed a certain size are readily visualized (*Serkova et al., 2021*). SPECT and PET can detect specific cell populations based on the use of radiolabeled ligands that recognize them, but lack single-cell resolution. The most widely used radioisotopes for SPECT include $^{123}$I and $^{99}$Tc, with half-lives of ~13 hr and ~6 hr, respectively (*Adak et al., 2012*). Radiolabeled immunoglobulins used as imaging agents show excellent specificity, but their molecular mass (~150 kDa) impedes efficient tissue penetration and imposes a long circulatory half-life. When using immunoglobulin-based imaging agents for PET, the use of long-lived radioisotopes such as $^{89}$Zr (t$_{1/2}$=~3.3 days) is, therefore, indicated.

The introduction of nanobodies for SPECT and PET overcomes many of the limitations of immunoglobulin-based imaging agents. The small size of nanobodies improves tissue penetration and drastically reduces the circulatory half-life of nanobody-based imaging agents. Free nanobodies

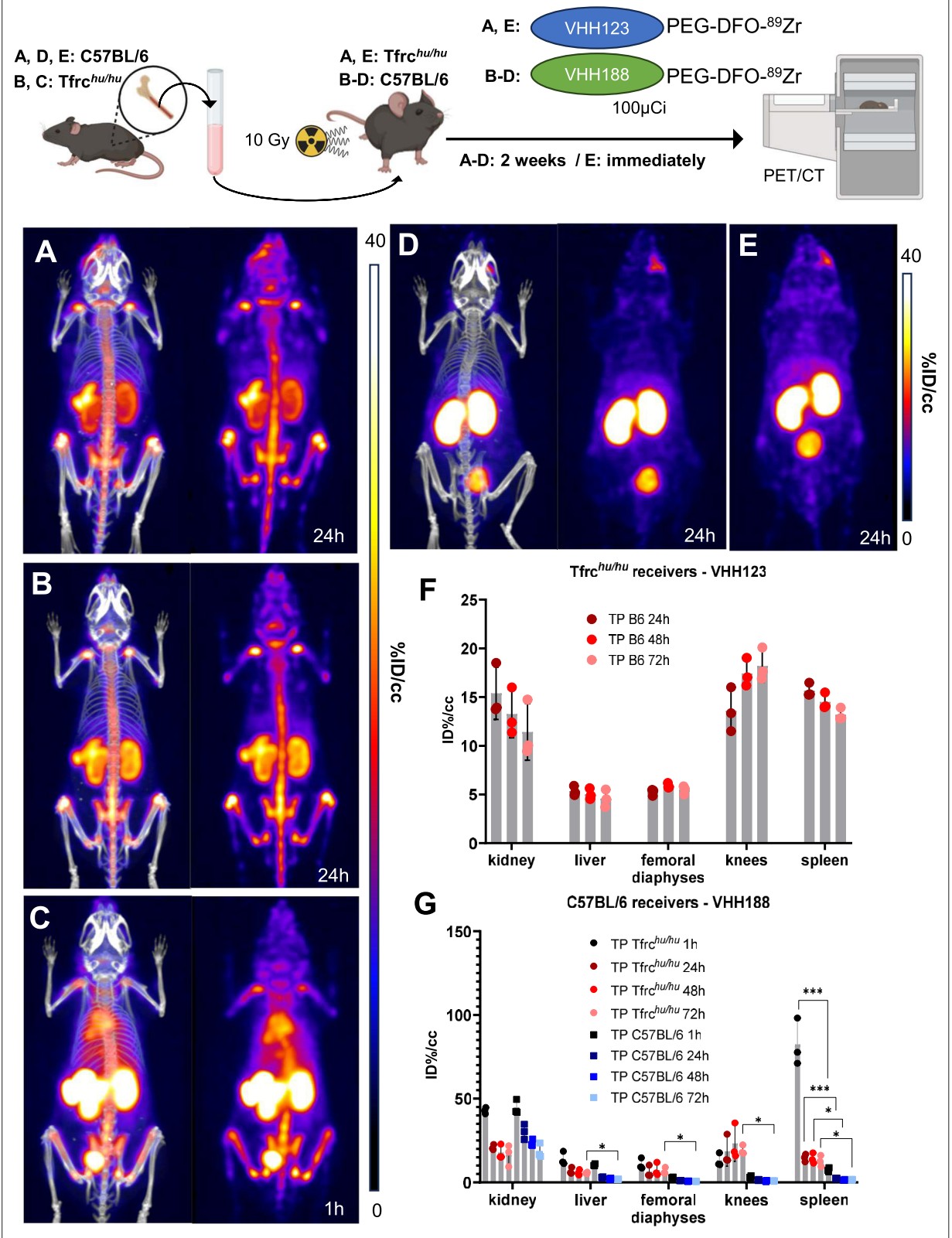

**Figure 6.** [89]Zr-labeled anti-TfR VHHs enable PET/CT tracking of transplanted bone marrow cells. Top: cartoon depicting the experimental procedure: bone marrow was harvested from the femur of C57BL/6 (**A, D, and E**) or Tfrc[hu/hu] (**B and C**) mice before transplantation into three lethally irradiated (10 Gy) Tfrc[hu/hu] (**A and E**) or C57BL/6 (**B, C, and D**) mice. These mice were then injected with 3.7 MBq (100 μCi) of $V_H$H123-PEG(20 kDa)-DFO-[89]Zr (**A and E**) or $V_H$H188-PEG(20 kDa)-DFO-[89]Zr (**B, C, and D**) immediately (**E**) or 2 weeks post-transplantation (**A, B, C, and D**) and positron emission tomography

*Figure 6 continued on next page*

*Figure 6 continued*

(PET)/CT images were acquired at several timepoints thereafter. Bottom: PET/CT maximum intensity projections (MIP) that were deemed the most representative of each condition are shown. Each panel comprises maximum intensity projection (MIP) overlayed with CT signal on the left, and PET MIP alone on the right. All repeats (1 repeat = 1 mouse) are from one experiment out of two replicated experiments. Each group has three mice. Groups D and E were control groups (specificity and timepoint, respectively). (**A**): MIP of one Tfrc*hu/hu* recipient mouse acquired 2 weeks after C57BL/6 bone marrow transplantation and 24 hr after radiotracer injection. (**B**): MIP of one C57BL/6 mouse acquired 2 weeks after Tfrc*hu/hu* bone marrow transplantation and 24 hr after radiotracer injection. (**C**) Same as B, but imaged 1 hr after radiotracer injection. (**D**) MIP of one C57BL/6 mouse acquired 2 weeks after C57BL/6 (isogenic) bone marrow transplantation and 24 hr after radiotracer injection. (**E**) MIP of one Tfrc*hu/hu* recipient mouse injected with radiotracer immediately after C57BL/6 bone marrow and imaged 24 hr thereafter. Two cohorts were set up separately to perform the PET/CT imaging immediately after bone marrow transplantation or two weeks thereafter. PET intensity scales are displayed on the right of each panel (%ID/cc). (**F**) ROI analysis of images acquired from mice as shown in panel A, and all repeats and imaging timepoints thereof. Error bars show SD. (**G**) Region of interest (ROI) analysis of images acquired from mice as shown in panels B, C, and D and all repeats and imaging timepoints thereof. Error bars show SD.

The online version of this article includes the following source data for figure 6:

**Source data 1.** PET/CT images of all repeats (mice) from the experiment shown in *Figure 6*.

are typically excreted via the kidneys and have a circulatory half-life of ~30 min, versus a half-life (in mice) of several days for intact immunoglobulins (*Oliveira et al., 2013*). This means that unbound nanobodies are rapidly cleared from the circulation to yield a much-improved signal-to-noise ratio. Conjugation of a PEG$_{20kDa}$ moiety further improves contrast by reducing somewhat the renal clearance rate of the nanobodies (*Rashidian et al., 2016*). Chemo-enzymatic methods for the modification of nanobodies allow their site-specific and reproducible modification with substituents of interest, including the installation of chelators for radiometals such as $^{89}$Zr and $^{64}$Cu, as well as click handles for site-specific covalent modification with $^{18}$F (*Harmand and Islam, 2021*).

For detection of tumors, xenografts in immunocompromised mice are widely used in combination with tumor-specific imaging agents. Typically, they are based on antibodies that recognize human-specific surface markers (*Freise and Wu, 2015*). Methods for the detection of tumors of mouse origin in their natural host are few and far between. It is for this reason that we explored the possibility of using a congenic pair of mice in combination with nanobodies that distinguish the congenic marker, in this case, the TfR. We show that a combination of anti-TfR nanobodies and mice that express either a wild-type (mouse) or a humanized TfR may be used to trace different cell types in vivo that express the TfR. To that end, we transferred hematopoietic bone marrow cells and B16.F10 melanoma cells into the respective mouse recipients. It is thus possible to track cells of mouse origin in humanized Tfrc*hu/hu* mice and humanized Tfrc*hu/hu* mouse bone marrow in C57BL/6 mice. Mice carrying Tfrc*hu/wt* fetuses showed a strong signal for VHH188 in the placenta, indicative of trans-endothelial transport of the imaging agent. Except for the TfR on syncytiotrophoblast-I, at 14 days of gestation, no other embryonic tissues showed any accumulation of label.

The combination of human and mouse-specific anti-TfR V$_H$Hs with mice that were engineered to express a TfR with the human TfR ectodomain constitutes a congenic pair of mice ideally suited for a variety of transplant experiments. The growth of tumors of mouse origin, when transplanted into Tfrc*hu/hu* mice, can be followed non-invasively, for example, in response to various treatments. Conversely, human patient-derived tumor xenografts (PDX models) can be followed upon transplantation into mice without a requirement for genetic modification of the transplant, even if no tumor-specific antibodies are available. Transplantation of C57BL/6-isolated tissues and tumor cells (-lines) in Tfrc*hu/hu* is straightforward, as both recipient and graft are of the same genetic background. The only required neo-epitope of the graft would be the ectodomain of the TfR, which we consider a negligible risk for immune rejection: the Tfrc*hu/hu* mouse expresses a chimeric TfR protein where only aminoacids 196–381 of the mouse TfR are replaced with the homologous aminoacids of the human TfR, that have a 74% pairwise identity by alignment using Clustal Omega (*Wouters et al., 2022*; *Madeira et al., 2024*). The lack of an obvious PET signal in the CNS when imaging mice may be due to the fact that proteins that traverse the BBB, such as the TfR nanobody conjugates, accumulate in the CNS at a concentration far lower (around 1–5%) than that in blood plasma (*Pothin et al., 2020*; *Meier et al., 2022*). Furthermore, $^{89}$Zr radioconjugates are prone to radioisotope release post-internalization, which is the first step of transcytosis required for plasma proteins to traverse the BBB. Finally, both the $^{89}$Zr and $^{64}$Cu radioconjugates were PEGylated, which may impede transcytosis. While the use of non-PEGylated $^{18}$F radioconjugates might circumvent these issues, their short circulatory and isotopic

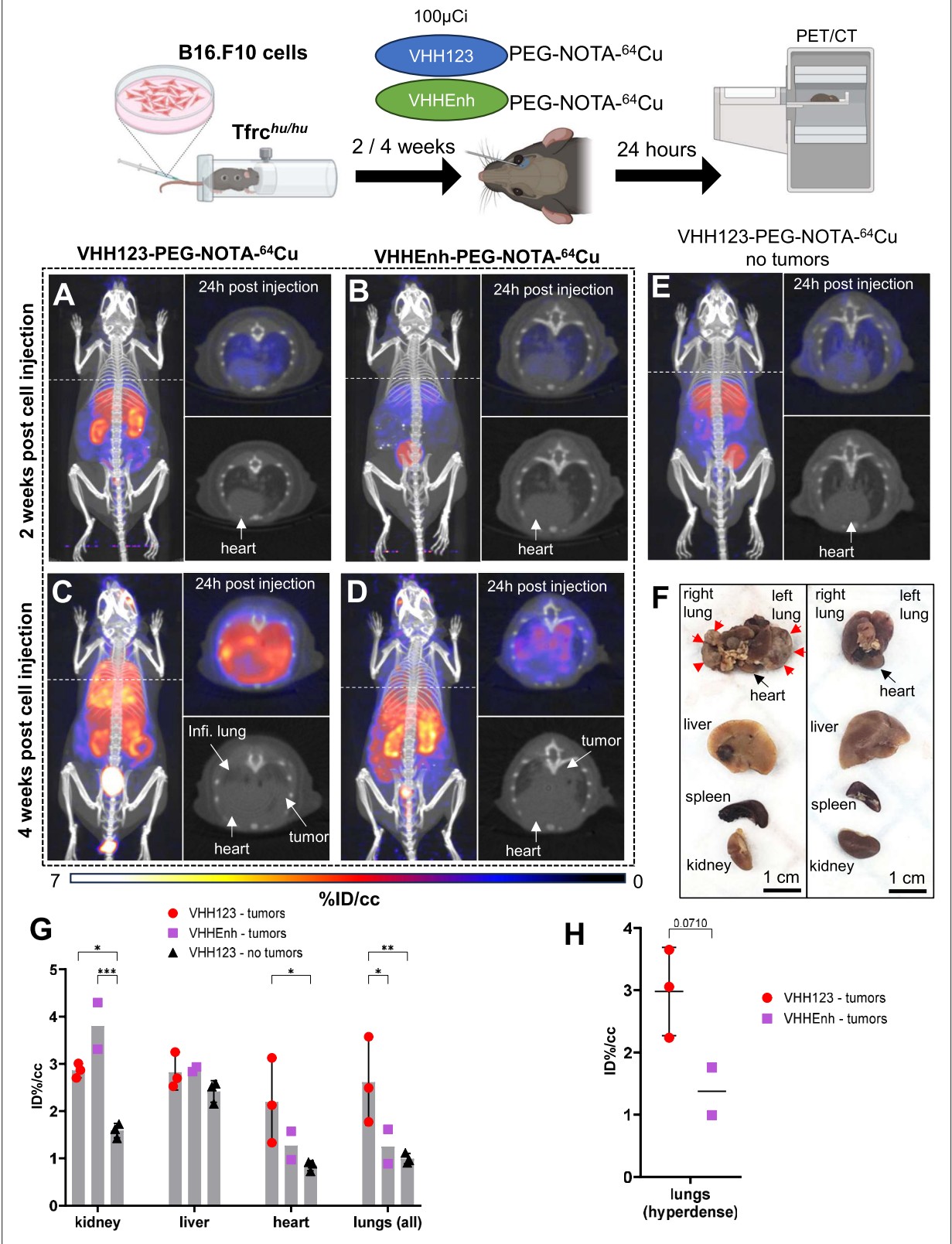

**Figure 7.** $^{64}$Cu-labeled VHH123 enables the detection of B16.F10 lung metastases by PET/CT. Top: cartoon depicting the experimental procedure: $5 \times 10^4$ B16.F10 mouse melanoma cells were transfused to Tfrc$^{hu/hu}$ mice by tail-vein injection. 2 and 4 weeks later, the mice were injected with 3.7 MBq (100 µCi) of either V$_H$H123-PEG(20 kDa)-NOTA-$^{64}$Cu or V$_H$HEnh-PEG(20 kDa)-NOTA-$^{64}$Cu radiotracers before PET/CT imaging. All repeats (1 repeat = 1 mouse) shown are from one experiment, performed once. (**A–E**) Maximum Intensity Projections (MIP) and lung traverse sections of one representative

*Figure 7 continued on next page*

*Figure 7 continued*

mouse out of three for each experimental condition at the 2 weeks timepoint post B16.F10 cell transfusion (except for E: no B16.F10 cells were injected). Images were acquired 24 hr post-radiotracer injection. Positron emission tomography (PET) intensity scale is displayed on the right (%ID/cc). (**A**) MIP of a Tfrc$^{hu/hu}$ mouse imaged with V$_H$H123-PEG-NOTA-$^{64}$Cu 2 weeks post B16.F10 cell transfusion. (**B**) MIP of a Tfrc$^{hu/hu}$ mouse imaged with V$_H$HEnh-PEG-NOTA-$^{64}$Cu 2 weeks post B16.F10 cell transfusion (non-specific V$_H$H control). (**C**) MIP of a Tfrc$^{hu/hu}$ mouse imaged with V$_H$H123-PEG-NOTA-$^{64}$Cu 4 weeks post B16.F10 cell transfusion. (**D**) MIP of a Tfrc$^{hu/hu}$ mouse imaged with V$_H$HEnh-PEG-NOTA-$^{64}$Cu 4 weeks post B16.F10 cell transfusion (non-specific VHH control). (**E**) MIP of a Tfrc$^{hu/hu}$ mouse imaged with V$_H$H123-PEG-NOTA-$^{64}$Cu that did not receive any tumor cells (no tumor control). (**F**) Photographs of dissected organs from Tfrc$^{hu/hu}$ mice euthanized at 4 weeks post B16.F10 cell infusion and 96 hr post radio-tracer injection (left) and from control mice that received no cells 96 hr post radio-tracer injection (right). RL: right lung, LL: left lung, H: heart, Li: liver, Sp: spleen, Ki: kidneys. Organs are from the same respective mice as shown in C. Red arrows delimit necrotic and hyperdense tumors growing out of the right and left lung. (**G**) Region of interest (ROI) analysis of images acquired from mice as shown in panels C, D, and E. Each dot represents the mean ID%/cc of a specific ROI for one mouse. Error bars show SD. No error bars are shown for the V$_H$HEnh – tumor group as n=2 (one mouse died before imaging). (**H**) ROI analysis of hyperdense lung tissue as visualized by CT on images acquired from mice as in panels C and D. Each point shows the mean ID%/cc of one mouse. Error bars show SD. No error bars are shown for the V$_H$HEnh – tumor group as n=2.

The online version of this article includes the following source data for figure 7:

**Source data 1.** PET/CT images of all repeats (mice) from the experiment shown in *Figure 7*.

---

half-life may not allow the visualization at adequate sensitivity of a signal in the CNS. Even so, gamma ray spectrometry on various organs from mice that received $^{89}$Zr-conjugated anti-TfR tracers confirmed accumulation of both nanobodies in the CNS of the appropriate genotype compared to control conditions (*Figure 3H*).

Release of free $^{89}$Zr and $^{64}$Cu by the nanobody conjugates was a surprising observation, unique to the anti-TfR V$_H$Hs. For no other $^{89}$Zr-labeled, PEGylated V$_H$H have we seen such a striking and rapid release of free $^{89}$Zr. The osteophilic properties of $^{89}$Zr have been well documented (*Raavé et al., 2019*; *Abou et al., 2011*) and confirmed in our hands (*Figure 3—figure supplement 2*). The signal generated by free $^{89}$Zr should, therefore, not be confused with the localization of anti-TfR nanobody to the spinal cord, bone marrow, or skeletal elements more generally. Careful interpretation is required when examining accumulation of tracer in bone marrow when using anti-TfR-$^{89}$Zr conjugates or any other internalizing $^{89}$Zr-labeled tracer. The fact that the observed release was unique to the anti-TfR V$_H$Hs, seen only in the presence of the appropriate target, may also be consistent with a mechanism that specifically targets the DFO-$^{89}$Zr chelate in compartments to which the TfR localizes. Perhaps release of Fe$^{+++}$ from Tf requires not only acidic pH but also the presence of some as yet unidentified co-factor that can act on the $^{89}$Zr-DFO chelate as well. Imaging experiments performed on pregnant C57BL/6 mice that carry Tfrc$^{hu/wt}$ embryos may shed further light on this question. Notwithstanding the very strong accumulation of label seen in the placenta, which we ascribe to the presence of the human-ectodomain modified TfR at the surface of the embryonic syncytiotrophoblast-I, we did not see any sign of release of free $^{89}$Zr. Either the TfR upon internalization into the syncytiotrophoblast-I is never exposed to the low pH responsible for $^{89}$Zr release in other tissues, but still sufficiently low to allow release of Fe$^{+++}$ from transferrin, or the hypothesized co-factor that mediates release of $^{89}$Zr from the DFO chelator is absent from the syncytiotrophoblast-I. We favor the former explanation because it would allow delivery of Fe$^{+++}$ to embryonic tissues and satisfy their demand for iron. Another possibility is that $^{89}$Zr is indeed released at the level of syncytiotrophoblasts-I, but would remain trapped at the interface between syncytiotrophoblasts-I and -II, as it would then not be able to penetrate the fetal circulation via ferroportins (*Sangkhae and Nemeth, 2019*). Release of free $^{89}$Zr from DFO after epitope-binding should also be a concern in the development of $^{89}$Zr-DFO radio-conjugates destined for clinical use.

The issue of the 'free $^{89}$Zr signal pattern,' defined as an intense signal originating from long bone extremities, vertebrae and coxal bone, induced by $^{89}$Zr release may be traded for an increased signal in the liver by shifting to the use of $^{64}$Cu as a PET isotope, which is attributed to the passive release of $^{64}$Cu from NOTA – a phenomenon that has been well characterized in the literature (*Dearling et al., 2011*; *Mirick et al., 1999*). The use of $^{18}$F-labeled anti-TfR V$_H$Hs conjugated through click chemistry avoids the complications of isotope release. However, $^{18}$F comes with its own drawbacks, such as its short half-life, necessitating less convenient and far more expensive synthetic routes for tracer production, and the typical accumulation of label seen in organs of elimination such as the gall bladder and gastrointestinal tract when using TCO-tetrazine click-chemistry (*Zhou et al., 2020*).

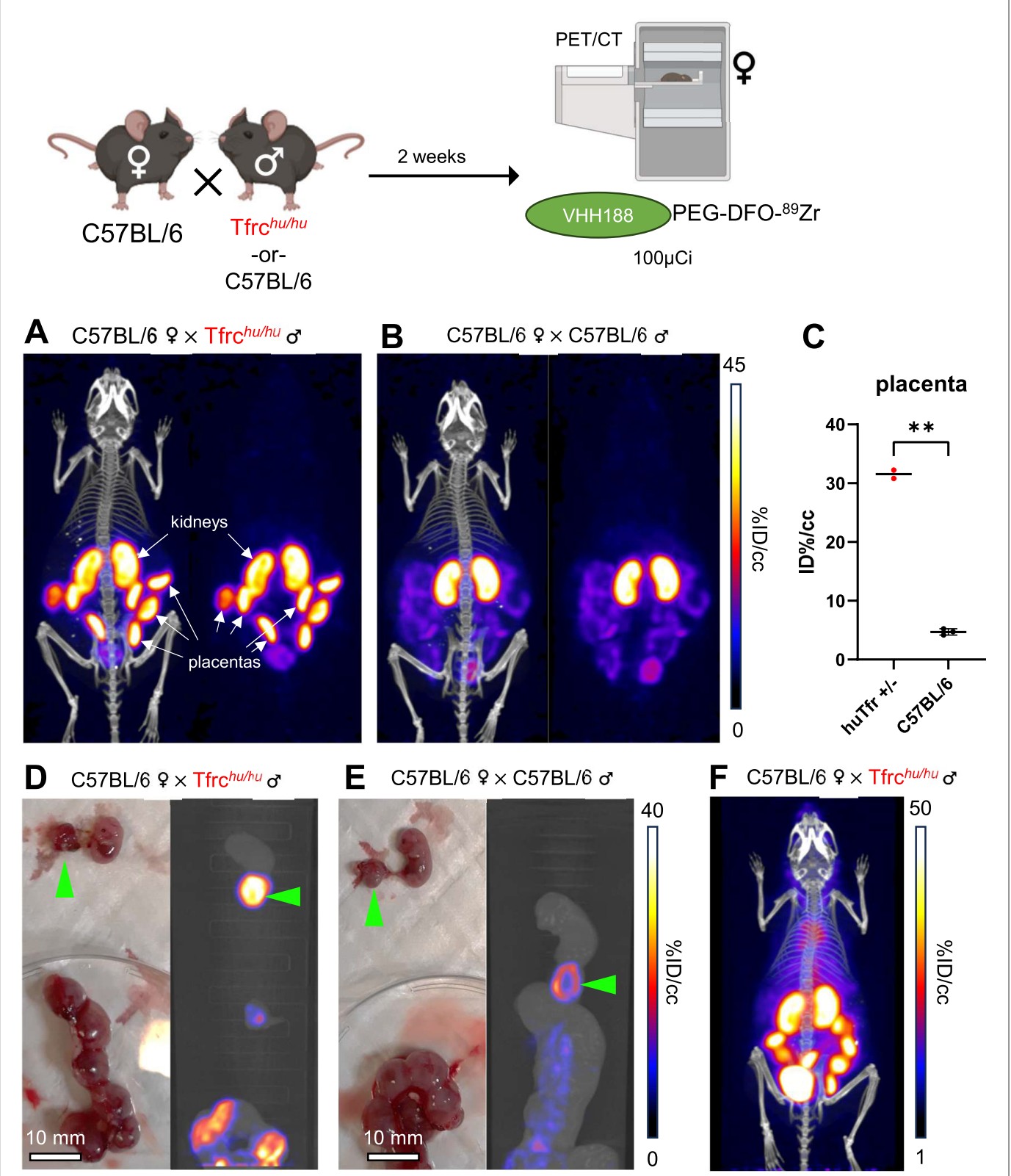

**Figure 8.** $^{89}$Zr-labeled VHH188 enables the detection of TfR at the blood-placenta barrier level by PET/CT. Top: cartoon depicting the experimental procedure: 8–12 week-old C57BL/6 females were mated in pairs with a single 12-week-old Tfrc$^{hu/hu}$ or C57BL/6 male. 2 weeks post-fertilization (confirmed by observing vaginal plugs), the females are injected retro-orbitally with 3.7 MBq (100 µCi) of V$_H$H188-PEG(20 kDa)-DFO-$^{89}$Zr. Experiment performed twice, all repeats included (1 repeat = 1 mouse). (**A**) PET/CT (left) and PET (right) maximum Intensity Projections (MIP) of one female carrying Tfrc$^{hu/wt}$

*Figure 8 continued on next page*

*Figure 8 continued*

embryos, imaged 24 hr after injection of the radiotracer. (**B**) PET/CT (left) and PET (right) MIP of one female carrying C57BL/6 wild-type embryos (control group), imaged 24 hr after injection of the radiotracer. Positron emission tomography (PET) intensity scale for A and B is displayed on the right (%ID/cc).(**C**) Region of interest (ROI) analysis of images acquired from mice as shown in panels A and B. Each dot represents the mean ID%/cc of the placenta for one mouse. Error bars show SD. No error bars are shown for the Tfrc*hu/wt* condition as n=2 (2 out of 4 females had plugs 2 weeks prior but were not gestating at time of imaging). (**D**) Left side panel: photograph of dissected embryos from the euthanized female carrying Tfrc*hu/wt* embryos, 72 hr post radiotracer injection. Right side panel: medial PET/CT section of a 50 mL tube containing the dissected embryos as shown on the left panel. The green arrows highlight the placenta of one embryo. (**E**) Left side panel: photograph of dissected embryos from the euthanized female carrying wild-type C57BL/6 embryos, 72 hr post radiotracer injection. Right side panel: medial PET/CT section of a 50 mL tube containing the dissected embryos as shown on the left panel. The green arrows highlight the placenta of one embryo. PET intensity scale for D and E is displayed on the right (%ID/cc). (**F**) PET/CT MIP of one female carrying Tfrc*hu/wt* embryos, imaged 1 hr after injection of the radiotracer. PET intensity scale is displayed on the right (%ID/cc). Experiment performed twice. Total of n=2 mice in B6 × Tfrc*hu/hu* group and n=3 in B6 × B6 group.

The online version of this article includes the following source data for figure 8:

**Source data 1.** PET/CT images of all repeats (mice) from the experiment shown in *Figure 8*.

In conclusion, this work demonstrates that the ubiquitously expressed TfR may be used as a cell marker to track virtually any cell type of choice in vivo, provided they are transferred to a mouse model that expresses a different isoform of TfR. The Tfrc*hu/wt* mouse model has been deposited at the Jackson Laboratories and is, therefore, easily accessible (strain number 038212). Production and sortagging of nanobodies to generate PET tracers is a relatively simple process. These nanobodies and the Tfrc*hu/hu* model may thus benefit the field of in vivo imaging.

# Materials and methods

## Key resources table

| Reagent type (species) or resource | Designation | Source or reference | Identifiers | Additional information |
|---|---|---|---|---|
| Strain, strain background (*Mus musculus*) | C57BL/6 J | The Jackson Laboratory | 000664; RRID:MGI:2159769 | |
| Strain, strain background (*Mus musculus*) | C57BL/6-Tfrc*tm1(TFRC)Bdes*/J | The Jackson Laboratory; *Wouters et al., 2022* | 038212; hAPI KI; Tfrc*hu/hu*; RRID:IMSR_JAX:038212 | Strain donated to Jackson Labs by Dr. M. Dewilde and colleagues |
| Peptide, recombinant protein | Anti-mouse transferrin receptor nanobody | *Wouters et al., 2020* | V$_H$H123; Nb63 | |
| Peptide, recombinant protein | Anti-human transferrin receptor nanobody | *Wouters et al., 2022* | V$_H$H188; Nb188 | |
| Cell line (*Mus musculus*) | B16-F10 | ATCC | B16-F10; CRL-6475; RRID:CVCL_0159 | |

## Production of nanobodies

cDNAs encoding V$_H$H123 (anti-mouse TfR nanobody) and V$_H$H188 (anti-human TfR nanobody) were cloned into a pHEN6 plasmid backbone that encodes the 'GGLPETGGHHHHHH' sortase A motif and histidine tag at the C-terminus of the expressed construct. WK6 *E. coli* were transformed with each plasmid vector and grown to saturation at 37 °C in Terrific Broth (Millipore Sigma) prior to induction with 1 mM IPTG and continued incubation overnight at 16 °C. Extraction of each V$_H$H was performed by osmotic shock as described (*Ingram and Dougan, 2025*). Purification of each V$_H$H was achieved through Ni$^{2+}$-NTA affinity chromatography followed by size-exclusion FPLC (Superdex 16/600 75 pg, Cytiva).

## Sortase A-mediated conjugation

10–30 µM of V$_H$H-GGLPETGG-His$_6$ were incubated overnight at 8 °C in the presence of 500 µM – 1 mM GGG-nucleophile, 30 µM of penta-mutant Sortase A-His$_6$ (produced in-house as previously described *Shi et al., 2014*, Addgene #51140), 2 mM CaCl$_2$ in 1 mL total volume of PBS. 300 µL Ni$^{2+}$-NTA beads were then added to the reaction to capture Sortase A and unreacted V$_H$H. The unbound fraction was

desalted on a gravity-fed PD-10 size-exclusion column (Cytiva) to separate the $V_HH$-conjugate from the excess of free GGG-nucleophile. Conjugation of each $V_HH$ was monitored by SDS-PAGE of each individual step of the reaction and by LC/MS of the purified product (QDa, Waters). Yields of sortase-mediated conjugations were >75% or higher (total output protein mass vs. total input protein mass – measured by A280 using a NanoDrop, Thermo Fisher).

## Immunoprecipitation

Conjugation of $V_HH123$ or $V_HH188$ to biotin was performed by Sortase A conjugation and GGG-biotin as the nucleophile, as described above. The biotin-conjugated $V_HHs$ were then incubated with 1 mg of streptavidin-coated paramagnetic beads (streptavidin Dynabeads T1, Thermo Fisher) for 1 hr at 4 °C in PBS. Unbound $V_HH$-biotin was removed by washing the beads three times in lysis buffer: 1% NP-40, 150 mM NaCl, 20 mM Tris-HCL pH 7.4, Halt Protease Inhibitor 1 X (Thermo Fisher). HEK293 and B16. F10 cells were grown to confluency before detaching using Versene solution. Cells were washed in PBS to remove excess medium and FBS before lysis in 100 μL lysis buffer/$1×10^6$ cells. Lysates were incubated on a rotator for 30 min at 4 °C before pelleting debris by centrifugation at 21,130 g for 5 min at 4 °C on a benchtop centrifuge. Supernatants were then pre-cleared with 100 μg/300 μL of lysate of streptavidin Dynabeads for 1 hr at 4 °C. The unbound fraction was removed after placing the tube on a magnetic rack, and incubated with 1 mg of $V_HH123$-biotin or $V_HH188$-biotin pre-coated Streptavidin Dynabeads (see above) overnight at 4 °C. Beads were then washed five times in lysis buffer and then 2 times in PBS. Elution was done by boiling the beads in SDS-PAGE sample buffer. Eluted proteins were resolved by SDS-PAGE.

## LC/MS/MS analysis

Sample preparation and analyses were performed by the Taplin Mass Spectrometry Core at Harvard Medical School. Excised SDS-PAGE gel sections were cut into approximately 1 mm$^3$ pieces. Gel pieces were then subjected to a modified in-gel trypsin digestion procedure (*Shevchenko et al., 1996*). Gel pieces were washed and dehydrated with acetonitrile for 10 min followed by removal of acetonitrile and lyophilization in a speed-vac. Gel pieces were then rehydrated with 50 mM ammonium bicarbonate solution containing 12.5 ng/μl modified sequencing-grade trypsin (Promega, Madison, WI) at 4 °C. After 45 min., the trypsin solution was removed and replaced with sufficient 50 mM ammonium bicarbonate solution to cover the gel pieces. Samples were then placed in a 37 °C room overnight. Peptides were recovered by removing the ammonium bicarbonate solution, followed by one wash with a solution containing 50% acetonitrile and 1% formic acid. The extracts were combined and dried in a speed-vac (~1 hr). Samples were reconstituted in 5–10 μl of HPLC solvent A (2.5% acetonitrile, 0.1% formic acid) and were applied to a nano-scale C18 reverse-phase HPLC capillary column. Peptides were eluted with increasing concentrations of solvent B (97.5% acetonitrile, 0.1% formic acid) and were subjected upon elution to electrospray ionization and then injected into a Velos Orbitrap Pro ion-trap mass spectrometer (Thermo Fisher Scientific, Waltham, MA). Peptides were detected, isolated, and fragmented to produce a tandem mass spectrum of specific fragment ions for each peptide. Peptide sequences (and hence protein identity) were determined by matching protein databases with the acquired fragmentation pattern by the software program, Sequest (Thermo Fisher Scientific, Waltham, MA) (*Eng et al., 1994*). All databases include a reversed version of all the sequences and the data was filtered to a one and two percent peptide false discovery rate. Subcellular localization annotation was retrieved by matching the Uniprot entry number of each protein with its Uniprot subcellular localization annotation, using the CellWhere database (*Zhu et al., 2015*).

## $^{35}$S-Cysteine/Methionine labeling of cells

HEK 293T and B16.F10 cells were grown to 80% confluency in 2x T75 plates each before careful washing 2 x with Cys/Met-free DMEM media with 10% dialyzed FBS. Cells were then starved in Met/Cys-free medium at 37 °C for 30 min with dialyzed FBS before replacing the medium with $^{35}$S-Cys/Met enriched medium (11 μCi/μL, EasyTag EXPRESS35S Protein Labeling Mix, Perkin Elmer) for 5 hr at 37 °C. $^{35}$S-labeled cells were then processed for immunoprecipitation as described above.

## Flow cytometry

A dilution range of different $V_HH$ concentrations prepared in PBS 2% FBS, were incubated with 0.1 million CHO cells overexpressing either the human or mouse TfR for 30 min at 4 °C. The binding of $V_HHs$ was next followed by a 30 min incubation at 4 °C with an anti-FLAG-iF647 antibody (A01811, Genscript, Piscataway, NJ, USA), diluted 1:500. Dead cells were stained with the viability dye eFluor780 (1:2000; 65-0865-14, Thermo Fisher Scientific, Waltham, MA, USA) for 30 min at 4 °C. Flp-In-CHO cells, used as unstained control and single stain controls, were used to determine the cutoff point between background fluorescence and positive populations. UltraComp eBeads Compensation Beads were used (01-2222-42, Thermo Fisher Scientific, Waltham, MA, USA) to generate single stain controls for the anti-FLAG-iF647 antibody. The data was acquired by using an Attune Nxt flow cytometer (Thermo Fisher Scientific, Waltham, MA, USA) and analyzed by FCS Express 7 Research Edition.

## Mice

C57BL/6 mice were purchased from Jackson Laboratories (strain 000664) and Tfrc$^{hu/hu}$ mice were provided by the groups of M. Dewilde and B. De Strooper (VIB and KU Leuven, Belgium) and bred in-house. The Tfrc$^{hu/hu}$ strain has been deposited at Jackson Laboratories (strain 038212). Mice were housed and handled according to the institution's IACUC policy #00001880 – with a 12 hr day/night cycle, min/max temperature of 20/23.3 °C, and water/food access ad libitum. Unless otherwise specified in figure legends, mice used in the experiments were between 8–12 weeks old. The number of mice per experiment was limited by the throughput of the PET/CT imager (~1 mouse/15 min), and so we aimed to have a minimum of 3 mice per condition/group in our experiments. Both male and female mice were included in our experimental groups, at an approximate 50/50 percent ratio. No blinding, randomization, strategy to minimize confounders, or inclusion/exclusion criteria were established. The outcome of the experiment was defined as the acquisition of the PET/CT images of good quality. Once acquisition was done, the mice were euthanized. For the B16F10 tumor experiment, humane endpoints were set up to prevent animal suffering: in case of weight loss >20% or other signs of distress, a mouse is euthanized immediately without any imaging being performed. ARRIVE v2.0 checklist submitted alongside manuscript.

## Transplantation of bone marrow

Femora and tibiae from 6 to 8 week-old C57BL/6 mice were flushed using a syringe equipped with a 23 G needle to harvest bone marrow in Iscove's Modified Dulbecco's Medium (IMDM). Cells were pelleted and resuspended in PBS, re-pelleted and then resuspended in 5 mL RBC lysis buffer (15 mM $NH_4Cl$, 1 mM $KHCO_3$, 1 µM disodium EDTA in water) and left for 10 min at room temperature. Cells were then washed twice in PBS to remove lysed RBCs prior to injection of $1.5 \times 10^6$ cells into a lethally irradiated (2x5 Gy, 4 hr apart) recipient Tfrc$^{hu/hu}$ mouse. Recipient mice were kept in individual restraining chambers during irradiation to maintain a uniform total body irradiation.

## Implantation of B16.F10 lung metastases

B16.F10 melanoma cells were grown to confluency, then detached with Versene solution 1 X, washed twice, and resuspended in sterile PBS at a concentration of $2.5 \times 10^5$ cell/mL. $5 \times 10^4$ cells were transfused *i.v.* by tail-vein injection in Tfrc$^{hu/hu}$ mice. Lung metastases began to appear ~ 3–4 weeks later, as confirmed by CT imaging of the lung and necropsy at the end of the experiment. Control mice that beared no tumors received PBS instead by tail-vein injection.

## Radioactive conjugate preparation

Conjugation to $^{89}Zr$ was based on previously published methods (*Rashidian et al., 2017*). $V_HHs$ were conjugated to GGG-Deferoxamine (DFO)-Azide by Sortase A transpeptidation (see above). The $V_HH$-DFO-Azide conjugate was then incubated with a fivefold molar excess of polyethylene-glycol$_{20kDa}$-dibenzylcyclooctyne (PEG$_{20kDa}$-DBCO) overnight at 4 °C on a shaker in PBS (pre-treated with Chelex beads in order to remove divalent cations) in order to generate $V_HH$-PEG$_{20kDa}$-DFO through click-chemistry. Completion of the reaction was assessed by SDS-PAGE. For radio-labeling, a stock solution of 129.5 MBq (3.5 mCi) of $^{89}Zr^{4+}$ in a 1 M oxalate solution (purchased from the Madison-Wisconsin University Cyclotron Lab) was adjusted to a pH of 6.8–7.5 with a 75% (vol/vol) of 2 M $Na_2CO_3$ and 400% (vol/vol) of 0.5 M HEPES buffer, pH 7.5. 37 MBq (1mCi) of pH-adjusted $^{89}Zr$ was then mixed with

100 µg of V$_H$H-PEG$_{20kDa}$-DFO for 1 hr at room temperature, followed by removal of unbound $^{89}$Zr using a PD-10 gravity desalting column (Cytiva) pre-equilibrated with Chelex-treated PBS. The column was eluted in fractions of 600 µL, the activity of which was measured using a dose calibrator (AtomLab 500, Biodex). The fraction corresponding to the peak activity (typically fraction 6 with an activity of typically 37 kBq/µL (1 µCi/µL)) was used for injection. The free $^{89}$Zr remaining in the desalting column was typically <10% of input: 37 kBq (100 µCi) suggesting a radioelement chelation efficiency of ~90%. For conjugation of $^{64}$Cu, GGG-NOTA-Azide was conjugated to the V$_H$H through sortase A transpeptidation (NOTA: 2,2',2"-(1,4,7-triazacyclononane-1,4,7-triyl)triacetic acid). The resulting conjugate was then further conjugated to PEG$_{20kDa}$ by click chemistry as described above. For radio-labeling, a stock solution of 1.48 GBq (40 mCi) of $^{64}$CuCl$_2$ (purchased from the Madison Wisconsin University Cyclotron Lab) was mixed with 150 µg of V$_H$H conjugate in PBS for 1 hr at room temperature on a shaker. Unbound $^{64}$Cu was removed from the mixture by passage onto a PD-10 gravity-fed desalting column (Cytiva) pre-equilibrated with PBS. The elution of the column, peak activity measurements, and injection doses were the same as with $^{89}$Zr, as described above. $^{18}$F-based V$_H$H conjugates were ready for injection post click-chemistry of the tetrazine-conjugated V$_H$Hs with $^{18}$F-TCO at the Molecular Cancer Imaging Facility at Dana-Farber Cancer Institute, Boston, M.A. (*Figure 2* and *Figure 2—figure supplements 1 and 2*). Synthesis of $^{18}$F-TCO is described below. In short, V$_H$H123-tetrazine (25 µL, 230 µM) was diluted with 668 µL of 1 x PBS. To this solution, 754.8 MBq (20.4 mCi) of $^{18}$F-TCO were added in 37 µL of EtOH. The reaction mixture was placed on a Thermomixer at 300 rpm at 25 °C. V$_H$H188-tetrazine (21 µL, 275 µM) was diluted with 843 µL of 1 x PBS and reacted with 943.5 MBq (25.5 mCi) of $^{18}$F-TCO in 45 µL of EtOH. Progress of click reactions was monitored by spotting iTLC-SG strip (Agilent, SGI0001) with 0.5 µL of reaction mixture at 5 and 10 min until about 20% of clicked product was detected in the mixture (10 min). For purification, 200 µL of TCO-agarose slurry (50% slurry in 20% EtOH, Click Chemistry Tools, 1198–5) was added to the top of a pre-equilibrated PD-10 column. The column was then washed with 30 mL of sterile 1 x PBS. Each $^{18}$F-TCO-tetrazine-V$_H$H click reaction mixture was loaded onto a column. When the solution reached bed level, 1.6 mL of 1 x PBS was added, and the $^{18}$F-radiolabeled V$_H$H was then eluted with 2x1 mL fractions of PBS. For $^{18}$F-V$_H$H123, the first 1 mL fraction contained 13.98 MBq (378 µCi) of product and the second 1 mL fraction measured 82.88 MBq (2.24 mCi). The first fraction was discarded, and after checking pH and radiochemical purity of the second fraction (iTLC-SG, *Figure 2—figure supplement 5*), the final formulation contained 78.07 MBq (2.11 mCi) of $^{18}$F-V$_H$H123 in 0.95 mL of 1 x PBS. For $^{18}$F-V$_H$H188, the first 1 mL fraction contained 30.97 MBq (837 µCi) of product and the second 1 mL fraction measured 138.38 MBq (3.74 mCi). The first fraction was discarded, and after checking pH and radiochemical purity of the second fraction (iTLC-SG, *Figure 2—figure supplement 5*), the final formulation contained 128.02 MBq (3.46 mCi) of $^{18}$F-V$_H$H188 in 0.91 mL in 1 x PBS. Each were used immediately after synthesis for radio-imaging. All final preparations of conjugates were confirmed to be ~pH 7.4 by testing with pH paper.

## PET/CT imaging

Mice were anaesthetized using 2.0% isoflurane in O$_2$ at a flow rate of ~1 liter per minute. For all radiotracers, 1.85–3.7 MBq (50–100 µCi) of radiotracer was injected retro-orbitally (typically in a 50–100 µL volume, depending on final activity/mL of radiotracer). PET/CT images were acquired using a G8 PET/CT machine (Sofie biotech – Perkin Elmer) with a 10 min PET signal detection window at several timepoints post-injection: 1–2 hr, 12 hr, 24 hr, then every 24 hr until 96 hr post-injection or until the radioisotope had decayed. Each PET acquisition was followed by a 1.5 min CT scan. Raw acquired images were processed by the manufacturer's automatic image reconstruction software to generate DICOM files. Images were then visualized, rendered, and analyzed using VivoQuant 3.5 software, patch 2 (Invicro). For ID%/cc, images were produced using ID%/g scales that were converted to ID%/cc by postulating that 1 mL of tissue = 1 g. Mice were kept anesthetized by continuous inhalation of 2.5% isoflurane during the acquisition of PET/CT images.

## Ex vivo measurement of activity

Mice injected with $^{89}$Zr-radiolabeled conjugates were euthanized by CO$_2$ inhalation. No capillary depletion was performed. Organs were harvested post-mortem and collected in pre-weighed 5 mL assay tubes. Each organ was weighed prior to gamma-counting using a Packard E5003 instrument. ID%/g (injected dose percentage per gram of tissue) for each sample was calculated using the following

formula: (activity of sample (MBq)/total injected activity (MBq))/sample weight (g)×100. For whole blood, sample weight was calculated by considering that 1 mL of whole blood = 1.06 g. Bone marrow was harvested by extensive flushing of harvested femurs with 4 mL total volume of PBS per femur. For comparing the activity of bone marrow vs emptied bone, weight of both sample types was considered as equal to 1, in order to generate a ID%/bone scale.

## Antibodies
Mouse monoclonal anti-TfR IgG (mouse and human cross-reactive), clone H68.4, Abcam catalog #ab269513.

## Cell lines
HEK 293 (originally from ATCC, ref#CRL-1573) and B16.F10 (originally from ATCC, ref#CRL-6475) cells were provided by the laboratory of Dr. Stephanie Dougan – Dana Farber Cancer Institute, Boston, MA, USA. Cells were tested and found negative for the presence of specific mouse pathogens and mycoplasma contamination.

## Labeling reagent synthesis
The synthetic routes of labeling reagents are shown in *Figure 2—figure supplement 3*, and their corresponding mass spectra are shown in *Figure 2—figure supplement 4*.

## Materials
Deferoxamine-DBCO (1) was purchased from Macrocyclics (cat. no. B-773). Azido-PEG3-Maleimide (2) was purchased from Vector Laboratories (cat. no. CCT-AZ107-100). Maleimido-mono-amide-NOTA (6) was obtained from Macrocyclics (cat. no. B-622). Methyltetrazine-Maleimide (8) was purchased from Conju-Probe (cat. no. CP-6608–25 mg). All the other chemical reagents and solvents were purchased from Sigma-Aldrich. The synthesis of the peptide GGG-PEG$_3$-Cys-PEG$_3$-Lys(azide) (4) and Gly-Gly-Gly-Cys (9) were synthesized as described (*Rashidian et al., 2017*; *Fang et al., 2016*). The molecular mass of the labeling reagents was determined using an LC-MS system (Waters QDa_Arc). All labeling reagents were purified by HPLC (Shimadzu) equipped with an XBridge BEH C18 OBD Prep Column (130 Å, 5 µm, 19 mm × 150 mm).

## Synthesis of GGG-DFO-N$_3$ (5)
To a solution of deferoxamine-DBCO (1) (25 mg, 0.029 mmol) in anhydrous DMSO (0.5 mL), azido-PEG$_3$-maleimide (2) (13 mg, 0.035 mmol) dissolved in anhydrous DMSO (0.5 mL) was added and stirred for 1 hr at room temperature. The reaction mixture was directly injected into HPLC (0–100% acetonitrile in H$_2$O containing 0.1% TFA), yielding intermediate (3) as a white powder (29 mg, 82%). LC-MS: [M+H]$^+$ = 1217.5.

A mixture of intermediate (3) (29 mg, 0.024 mmol) and peptide GGG-PEG$_3$-Cys-PEG$_3$-Lys(azide) (4) (41 mg, 0.048 mmol), dissolved in DMSO (1.5 mL), was stirred overnight at room temperature. The reaction mixture was then directly injected into HPLC (0–100% acetonitrile in H$_2$O containing 0.1% TFA), yielding GGG-DFO-N$_3$ (5) as a colorless oil (32 mg, 64%). LC-MS: [M+2 H]$^{2+}$ = 1035.6.

## Synthesis of GGG-NOTA-N$_3$
A mixture of GGG-PEG$_3$-Cys-PEG$_3$-Lys(azide) (4) (78 mg, 0.092 mmol) and maleimido-mono-amide-NOTA (6) (25 mg, 0.046 mmol), dissolved in DMSO (2 mL), was stirred overnight at room temperature. The reaction mixture was then directly injected into HPLC (0–100% acetonitrile in H$_2$O containing 0.1% TFA), yielding GGG-NOTA-N$_3$ (7) as a colorless oil (41 mg, 70%). LC-MS: [M+2 H]$^{2+}$ = 639.2.

## Synthesis of GGG-tetrazine
A mixture of methyltetrazine-maleimide (8) (25 mg, 0.071 mmol) and Gly-Gly-Gly-Cys (9) (62 mg, 0.213 mmol), dissolved in DMSO (1.5 mL), was stirred overnight at room temperature. The reaction mixture was then directly injected into HPLC (0–100% acetonitrile in H$_2$O containing 0.1% TFA), yielding GGG-tetrazine (10) as a red powder (23 mg, 50%). LC-MS: [M+H]$^+$ = 645.0.

## Radiosynthetic method for generating [18]F-TCO

Approximately 3000 mCi of [18 F]fluoride was produced on a GE PETtrace 800 cyclotron and delivered to the 18 F target delivery vial of a GE FX2 N radiosynthesis module. The irradiated target water was passed through a pre-conditioned Sep-PAK Light QMA Cartridge to trap [18 F]fluoride, followed by elution from the cartridge into reactor 1 using 2.5 mg K2CO2 and 18 mg Cryptand 222 in 0.1 mL of $H_2O$ (HyClone) and 0.9 mL acetonitrile (HPLC). The contents of reactor 1 were dried via azeotropic distillation using a combination of heating, helium flow, and vacuum. An additional 1 mL of acetonitrile (anhydrous) was introduced to reactor 1 and the azeotropic drying process was repeated. The reactor temperature was then reduced to 40 °C, followed by the introduction of TCO-nosylate (4 mg, Peptech) in 0.9 mL acetonitrile (anhydrous). The reactor was then sealed and heated to 75 °C and allowed to react with stirring for 10 min. Temperature of the reactor was cooled down to 40 °C and the reaction was quenched with the addition of 5 mL of sodium ascorbate solution (6.5 mg/mL). The reaction mixture was filtered through an Alumina N Sep-Pak cartridge (pre-conditioned with 15 mL of H2O) and loaded onto a 5 mL HPLC Loop which had been previously filled with semi-preparative HPLC mobile phase (80% acetonitrile, HPLC, in water, HPLC) to minimize injection of air onto the HPLC column. The crude mixture was then injected on a C18 semi-preparative HPLC column (flow rate = 3.5 mL/min). The 18F-TCO product peak (~8.5–9.2 min – *Figure 2—figure supplement 6*) was collected into a round bottom flask containing 20 mL sterile water. The contents of the round bottom flask were then passed through a C18 Plus Light Sep-PAK cartridge, where 18F-TCO was trapped. The cartridge was then washed with 5 mL water (USP, sterile for irrigation), followed by elution of 18F-TCO to the FX2 N product vial using 0.5 mL ethanol. 291 mCi of 18F-TCO was obtained in 0.42 g of EtOH for generation of VHH123-[18]F; 237 mCi of 18-TCO was obtained in 0.34 g of EtOH for generation of VHH188-[18]F. Methods for click-chemistry of 18F-TCO onto VHH-tetrazine conjugates is described above.

## Autoradiography

To perform autoradiography, after running, the SDS-PAGE gel was soaked in DMSO for 30 min, twice, to remove all aqueous solutions. Gel was then incubated in DMSO with 20% w/v 2,5-diphenyloxazole (PPO, insoluble in water) for 1 hr to allow PPO incorporation. Gel was then washed multiple times in water and dried using a gel dryer. Then, the gel was placed against an X-ray film in a cassette and stored at –80 °C for 24 hr before developing the film.

## Statistics

For every comparison of one condition between two populations, unpaired two-tailed t-tests using Welch's correction for non-equal standard deviations was performed. For every comparison of multiple conditions between two populations, unpaired two-tailed t-tests were performed, with Holm-Šídák's correction for multiple comparisons. For every comparison of one condition across three or more populations, one-way ANOVA with Tukey's correction for multiple comparisons was performed. For every comparison of two or more conditions across three or more populations, two-way ANOVA was performed, with Holm-Šídák's correction for multiple comparisons by considering all comparisons. All these analyses were done using GraphPad Prism v. 10.2.3. When considering experiments containing repeats at different timepoints – comparisons were done only between populations within the same timepoint. For all graphs: $p \leq 0.05$: *, $p < 0.01$: **, $p < 0.001$: ***, $p < 0.0001$: ****.

Experimental repeats are indicated in the figure legends. All repeats are biological repeats.

## Acknowledgements

The authors thank the Molecular Cancer Imaging Facility at Dana-Farber Cancer Institute, Boston, MA for the conjugation of [18]F to the VHH-tetrazine constructs and the Taplin Mass Spectrometry Core at Harvard Medical School, Boston, MA for the LC/MSMS analysis. TB acknowledges support of a Belgian American Educational Foundation post-doctoral fellowship and of a WBI. World fellowship from Wallonie-Bruxelles International. BDS and MD acknowledges a Grand Challenges grant from the Vlaams Instituut voor Biotechnologie (VIB), Ghent, Belgium. MD and TJ acknowledge Interne Fondsen KU Leuven/Internal Funds KU Leuven for its financial support. BDS is grateful for Methusalem support from the KU Leuven.

# Additional information

## Competing interests

Bart De Strooper: BDS and MD have submitted a PCT (Patent Cooperation Treaty) application related to anti-transferrin receptor nanobodies, distinct from those described in the present manuscript. BDS has been a consultant for Eli Lilly, Biogen, Janssen Pharmaceutica, Eisai, Muna Therapeutics and AbbVie and other companies, but not related to the current work. BDS is a scientific founder of Augustine Therapeutics and a scientific founder and stockholder of Muna Therapeutics. Maarten Dewilde: BDS and MD have submitted a PCT (Patent Cooperation Treaty) application related to anti-transferrin receptor nanobodies, distinct from those described in the present manuscript. The other authors declare that no competing interests exist.

## Funding

| Funder | Grant reference number | Author |
|---|---|---|
| Belgian American Educational Foundation | 2019-FELLOW-E044/PDF | Thomas Balligand |
| Wallonie-Bruxelles International | WBI.World SUB/2021/512532 | Thomas Balligand |
| Vlaams Instituut voor Biotechnologie | Grand Challenges GC01-C04 | Bart De Strooper Maarten Dewilde |
| KU Leuven | Interne Fondsen | Tom Jaspers Maarten Dewilde |
| KU Leuven | Methusalem | Bart De Strooper |

The funders had no role in study design, data collection and interpretation, or the decision to submit the work for publication.

## Author contributions

Thomas Balligand, Conceptualization, Investigation, Methodology, Writing – original draft, Writing – review and editing; Claire Carpenet, Sergi Olivé Palau, Mohammad Rashidian, Conceptualization, Investigation, Methodology, Writing – original draft; Tom Jaspers, Pavana Suresh, Xin Liu, Himadri Medhi, Yoon Ho Lee, Conceptualization, Investigation, Methodology; Bart De Strooper, Conceptualization, Supervision, Writing – original draft; Hidde L Ploegh, Maarten Dewilde, Conceptualization, Supervision, Investigation, Methodology, Writing – original draft, Writing – review and editing

## Author ORCIDs

Thomas Balligand ⓘ https://orcid.org/0000-0003-0156-8936
Claire Carpenet ⓘ https://orcid.org/0000-0002-7688-8574
Pavana Suresh ⓘ https://orcid.org/0000-0003-1597-6818
Xin Liu ⓘ https://orcid.org/0000-0002-2173-4201
Himadri Medhi ⓘ https://orcid.org/0000-0002-1757-6327
Bart De Strooper ⓘ https://orcid.org/0000-0001-5455-5819
Maarten Dewilde ⓘ https://orcid.org/0000-0002-3138-281X

## Ethics

This study was performed in strict accordance with the recommendations in the Guide for the Care and Use of Laboratory Animals of the National Institutes of Health. All of the animals were handled according to approved institutional animal care and use committee (IACUC) protocols (#00001880) of Boston Children's Hospital. All PET/CT imaging was performed under isoflurane anesthesia, and every effort was made to minimize suffering.

Reviewer #1 (Public review): https://doi.org/10.7554/eLife.104302.3.sa1
Reviewer #2 (Public review): https://doi.org/10.7554/eLife.104302.3.sa2
Author response https://doi.org/10.7554/eLife.104302.3.sa3

## Additional files

### Supplementary files
Supplementary file 1. Contains every peptide identified by LC/MS as in *Table 1*.

MDAR checklist

### Data availability
All data generated or analysed during this study are included in the manuscript and supporting files; source data files have been provided for Figures 1 and 2.

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
