## [Editor Report · eLife Assessment]

In this highly innovative study, Carpenet C et al explore the use of nanobody-based PET imaging to track proliferative cells after in vivo transplantation in mice, in a fully immunocompetent setting. The development of a unique set of PET tracers and mouse strains to track genetically-unmodified transplanted cells in vivo is an **important** novel asset that could potentially facilitate cell tracking in different research fields. The evidence provided is **compelling** as the new method proposed might facilitate overcoming certain limitations of alternative approaches, such as full sized immunoglobulins and small molecules.

---

## [Referee Report · Reviewer #1 (Public review)]

Summary:

The topic of nanobody-based PET imaging is important, and holds great potential for real-world applications since nanobodies have many advantages over full sized immunoglobulins and small molecules.

Strengths:

The submitted manuscript contains quite a bit of interesting data from a collaborative team of well-respected researchers. The authors are to be congratulated for presenting results that may not have turned out the way they had hoped, and doing so in a transparent fashion.

Weaknesses:

However, the manuscript could be considered to be a collection of exploratory findings rather than a complete and mature scientific exposition. Most of the sample sizes were 3 per group, which is fine for exploratory work, but insufficient to draw strong, statistically robust conclusions for definitive results.

Overall, the following specific limitations are noted as suggestions for future work:

(1) The authors used DFO, which is well known to leak Zr, rather than the current standard for 89Zr PET which is DFO* (DFO-star)

(2) The brain tissues were not capillary depleted, which limits interpretation. Capillary depletion, with quantitative assessment of the completion of the depletion process, is the standard in the field.

(3) The authors have not experimentally tested the hypothesis that the PEG adduct reduced BBB transcytosis.

(4) The results in Fig. 7 involving the placenta are interesting, but need confirmation using constructs with 18F labeling and without the PEG adduct.

(5) If this line of investigation were to be translated to humans, an important consideration would be the relative safety of 89Zr and 64Cu. It is likely to be quite a bit worse than for 18F, since the 89Zr and 64Cu have longer half-lives, dissociate from their chelators, and lodge in off-target tissues.

(6) A surprising and somewhat disappointing finding was the modest amount of BBB transcytosis. Clearly additional work will be needed before nanobody-based brain PET becomes feasible.

---

## [Referee Report · Reviewer #2 (Public review)]

Summary:

In this study the authors described a previously developed set of VHH-based PET tracers to track transplants (cancer cells, embryo's) in a murine immune-competent environment.

Strengths:

Unique set of PET tracer and mouse strain to track transplanted cells in vivo without genetic modification of the transplanted cells. This is a unique asset and a first-in-kind.

Weaknesses:

None

---

## [Author Response]

The following is the authors’ response to the original reviews.

**Reviewer #1 (Recommendations for the authors):**
Overall, the manuscript could be clearer and more beneficial to the readers with the following suggested revisions:(1) The abstract should include information on the comparative performance of 89Zr 64Cu and 18F labeled nanobodies, especially noting the challenges with DFO-89Zr and NOTA-64Cu.(2) The abstract should explicitly note the types of transplants assessed and the specific PET findings.(3) The abstract should note the negative results in terms of brain PET findings.

We thank reviewer 1 for these three suggestions. We have now included this information in the abstract.

(4) Based on the data shown in Fig. 1 and Table 1, it seems that the nanobodies bind to quite a few proteins other than TfR. This should be discussed frankly as a limitation.

The presence of multiple other bands and proteins identified by LC/MS in Figure 1 is typical for immunoprecipitation experiments, as performed under the conditions used: all proteins other than TfR that are identified in Table 1 are abundant cytoplasmic (cytoskeletal) and/or nuclear proteins. More rigorous washing would perhaps have removed some of these contaminants at the risk of losing some of the specific signal as well. We have added a comment to this effect. In an in vivo setting, this would be of minor concern, as these proteins would be inaccessible to our nanobodies. In fact, when VHH123 radioconjugates are injected in huTfr+/+ mice (or VHH188 in C57BL/6), we observe no specific signal – which supports this conclusion.

We therefore state: “We show that both V_H_Hs bind only to the appropriate TfR, with no obvious cross-reactivity to other surface-expressed proteins by immunoblot, LC/MSMS analysis of immunoprecipitates, SDS-PAGE of ^35^S-labelled proteins and flow cytometry (Fig 1;Table 1).”. We have added some clarification to make this clearer, and we also include the full LC/MSMS data tables are also added in supplemental materials, as supplementary Table 1. We have included subcellular localization information for each protein identified through LC/MSMS in Table 1 as well.

(5) Why did the authors use DFO, which is well known to leak Zr, rather than the current standard for 89Zr PET, DFO* (DFO-star)?

We used DFO rather than DFO-star for several reasons: (1) because we had already conducted and published numerous other studies using DFO-conjugated nanobodies and not observed any release of ^89^Zr, (2) commercially sourced clickchemistry enabled DFO-star (such as DFO*-DBCO) was not available at the time of the study.

(6) Figure 2B appears to show complex structures, more complex than just GGG-DFOazide, and GGG-NOTA-azide. This should be explained in detail.

We have added two supplemental figures and methods that recapitulate how we generated what we have termed as GGG-DFO-Azide and GGG-NOTA-Azide. We have updated the legend of Figure 2B.

(7) Why is there a double band in Suppl. Fig 9 for VHH123-NOTA-Azide?

Under optimal conditions, sortase A-mediated transpeptidation is efficient, resulting in the formation of a peptide bond between the C-terminally LPETG-tagged protein and the GGG-probe. However, extended reaction times or suboptimal concentrations of modified GGG-probes (which are often in limited supply) in the reaction mixture, allow hydrolysis of the sortase A-LPET-protein intermediate. The hydrolysis product can no longer participate in a sortase A reaction. This is what explains the doublet in the reaction used to generate VHH123-NOTA-N_3_ – the upper band is VHH123-NOTA-N_3_ and the lower band is the hydrolysis product. VHH123-LPET, is unable to react with PEG_20kDa_-DBCO (the lower band that appears at the same position of migration in the next lane on the gel). We noticed that an adjacent lane was mislabelled as ‘VHH188-NOTA-PEG_20kDa_’ when in fact it was ‘VHH123-NOTA-PEG_20kDa_’. This has been corrected.

The hydrolysis product, VHH123-LPET, has a short circulatory half-life and obviously lacks the PEG moiety as well as the chelator. It therefore cannot chelate ^64^Cu. Its presence should not interfere with PET imaging. Since all animals were injected with the same measured dose of ^64^Cu labeled-conjugate, the presence of an unlabeled TfRbinding competitor in the form of VHH123-LPET - at a << 1:1 molar ratio to the labelled nanobody – would be of no consequence.

(8) More details should be provided about the tetrazine-TCO click chemistry for 18F labeling.

We have added supplementary methods and figures that detail how ^18^F-TCO was generated. For the principle of TCO-tetrazine click-chemistry, a brief description was added in the text, as well as a reference to a review on the subject.

(9) For the data shown in Figure 3H, the authors should state whether the brain tissues were capillary depleted, and if so, how this was performed and how complete the procedure was.

No capillary depletion of the brain tissues was performed, as this was challenging to perform in compliance with the radiosafety protocols in place at our institution. We have updated the legend of figure 3H and methods to include this important detail. Whole blood gamma-counting did not show any obvious di erence of activity across the 4 groups in figure 3G (same mice as in figure 3H), which would go against the interpretation that activity di erences in the brain (figure 3H) are solely attributable to residual activity from blood in the capillaries.

(10) The authors should experimentally test the hypotheses that the PEG adduct reduced BBB transcytosis.

Reviewer 1 is correct to point out that we have not tested un-PEGylated conjugates of ^64^Cu and ^89^Zr with the anti-TfR nanobodies and we currently do not have the means to perform additional experiments. However, the ^18^F conjugates were not PEGylated, and these also fail to show any detectable signal in the CNS by PET/CT (see figure 4A). PEGylation alone cannot be the sole factor that limits transcytosis across the BBB.

(11) It was interesting to note that the Cu appears to dissociate from the NOTA chelator. The authors should provide more information about the kinetics of this process.

We have not tested the kinetics of dissociation between ^64^Cu and the NOTA conjugates in vitro, like we have done for ^89^Zr and DFO (supplemental figure 2), because previous work (see references 35 and 36 by Dearling JL and Mirick GR and colleagues) has shown that NOTA and other copper chelators tend to release free copper radioisotopes in the liver, a commonly reported artifact. We have also included a new set of images that show the biodistribution of VHH123-NOTA-^64^Cu in huTfR+/+ mice, where we still observe a substantial signal in the liver, indicating release of ^64^Cu from NOTA, in the absence of the anti-TfR VHH binding to its target. This was clearly not seen using the DFO-^89^Zr conjugates. Binding of the VHH to TfR, followed by internalization, appears to be required for the release of ^89^Zr from DFO, prompting us to investigate this phenomenon further.

(12) The authors should increase the sample size, and test two different radiolabels for the transplant imaging results (Figs. 5 and 6), since these seem to be the ones they feel are the most important, based on the title and abstract.

We agree with reviewer 1 that more repeats would increase the significance of our findings, but we unfortunately do not have the means of performing additional experiments at this time (the lab at Boston Children’s Hospital has closed as Dr. Ploegh has retired). We believe that the results are compelling and will be of use to the in vivo imaging community.

(13) Fig. 6G appears to show a false positive result for the kidney imaging. Is this real, or an artifact of small sample size?

We agree with reviewer 1 that the kidney signals in figure 6 are somewhat puzzling. The difference between the tumor-bearing mice that received VHH123 and VHHEnh conjugates is not significant – with the obvious caveat that the VHHEnh group is comprised of only 2 mice, so sample size may well be a factor here. If we compare the signals of the VHH123 conjugate in tumor-bearing mice vs. tumor-free mice, the VHH123 conjugates would have cleared much faster in the tumor-free mice over 24 hours (since no epitope is present for VHH123 to bind to), thus weakening the kidney signal observed after 24 hours. The same would be true for all the other tissues – except for the liver (where free ^64^Cu that leaks from NOTA accumulates). VHHEnh conjugates in tumor-bearing mice show a significant kidney signal – although no VHH123 target epitope is present in these mice. B16.F10 tumors at 4 weeks of growth tend to be necrotic and can passively retain any radiotracer – this generates the weak lung signal visible in Fig 6D – thus the radiotracer would clear at a slower rate than VHH123 conjugates in tumor-free mice giving a higher kidney signal at 24 hours.

No tumors were found in the kidneys post-necropsy. We attribute the differences in kidney signals to di erent kinetics of clearance of the radioconjugates. We have added this explanation to the results and discussion.

(14) Are the results shown in Fig. 7 generalizable? The authors should the constructs with 18F labeling and without the PEG adduct.

We agree with reviewer 1 that it would be very interesting to confirm these observations using 18F radioconjugates. The results should be generalizable, as the difference between signals can only be attributed to the presence of the recognized epitope in the placenta– which is in fact the only variable that differs between the two groups. At the time of conducting the study, we had not planned to perform the same experiments with 18F radioconjugates – partly because synthesis of 18F radioconjugates is more challenging (and costly) than the production of 89Zr-labeled nanobodies.

(15) The authors should discuss the relative safety of 89Zr and 64Cu. It is likely to be quite a bit worse than for 18F, since the 89Zr and 64Cu have longer half-lives, dissociate from their chelators, and lodge in off-target tissues. An alternative interpretation of the authors' data could be that 89Zr and 64Cu labeling in this context are unsuitable for the stated purposes of PET imaging. In this case, the key experiments shown in Figs. 5-7 should be repeated with the 18F labeled nanobody constructs.

Our vision was to o er a tool to the scientific community interested in in vivo tracking of cells in di erent preclinical disease models. The question of safety regarding 89Zr and 64Cu for clinical use was therefore not a factor we then considered. However, we have now included a section in the discussion about the potential safety issue of ^89^Zr release and bone accumulation in clinical settings, especially for radioconjugates that target an internalizing surface protein.

(16) The authors should remark on the somewhat surprisingly modest amount of BBB transcytosis in the discussion. What were the a inities of the nanobodies?

The a inities and binding kinetics of both nanobodies was described in a separate work that is referenced in the introduction (references 21 and 22 by Wouters Y and colleagues). Through other methods that rely on a highly sensitive bio-assay, it was shown that both VHH123 and VHH188 are capable of transcytosis: both nanobodies coupled to a neurotensin peptide induced a drop of temperature after i.v. injection in matching mouse strains (VHH123 in C57BL/6 and VHH188 in huTfr +/+). The lack of any compelling CNS signal by PET/CT is discussed in the manuscript.

(17) More details of the methods should be provided in the supplement.a. What was the source of the penta-mutant Sortase A-His6?

Sortase A pentamutant is produced in-house, by cytoplasmic expression in *E. coli* (BL21 strain), using a plasmid vector encoding a truncated and mutated version of Sortase A. References were added, as well as the Addgene repository number (51140).

b. What was the yield of the sortase reactions?

For small proteins, such as nanobodies/ V_H_Hs, we find that the yield of a sortase A reaction typically is > 75%. This is what we observed for all our conjugations. The methods section was updated to include this information.

c. What was the source of the GGG-Azide-DFO and GGG-Azide NOTA? Based on the structures shown in Fig. 2, these appear to be more complex that was noted in the text.

We have now detailed the synthesis of GGG-DFO-Azide and GGG-NOTA-Azide in the supplementary methods.

d. More details about the source and purity of the tetrazine and TCO labeling reagents should be provided.

We have included information on the synthesis of GGG-tetrazine in the supplementary methods. Concerning the synthesis of ^18^F-TCO, we have also included a detailed description of the compound in supplementary methods. The reaction between GGG-tetrazine and ^18^F-TCO is now further detailed in the manuscript.

e. The TCO-agarose slurry purification should be explained in more detail, and the results should be shown.

We have included a detailed procedure of how the TCO-agarose slurry purification was performed in the methods sections. We had already included the Radio-Thin Layer Chromatography QC data of the final VHH123-18F and VHH188-18F purifications in the supplementary figures – which are obtained immediately after TCOagarose slurry purification. The detailed yields of the TCO-agarose slurry purification in terms of activity of each collected fraction is now detailed in the methods section.

f. The CT parameters should be provided.

We have now added more information about the PET/CT imaging procedure in the methods section of the manuscript.

**Reviewer #2 (Recommendations for the authors):**
Authors should discuss the possibility of the TfR as a rejection antigen. Murine TfR is foreign for hTfR+/+ mice and vice versa.

We have not discussed this possibility, as we believe the risk of rejection of huTfR+ cells in moTfR+ mice (or vice versa) is negligible. The cells and mice are of the same genetic background – save for the coding region of ectodomain of the TfR (spanning amino acids ~194 to 390 of the full length TfR, which is 763 AA). The pairwise identity of both human and mouse TfR ectodomains is of 73% after alignment of both AA sequences using Clustal Omega. We agree that we cannot formally exclude the possibility of an immune rejection, and have now mentioned this possibility in the discussion.

Is there any clinical use of the anti-human TfR receptor PET tracer?

We do not currently envision an application for the anti-human TfR VHH in PET/CT in a clinical setting.

Why is the in vivo anti-mouse TfR uptake level in C57BL/6 mice consistently higher than the anti-human TfR receptor PET tracer in hTfR+/+ mice? Is this due to differences in characteristics of the VHH's (e.g. a inity, internalization properties), or rather due to a different biological behavior of the hTfR-transgene (e.g. reduced internalization properties)?

We indeed observed that VHH123 uptake and binding appears to be more robust than that of VHH188 to their respective targets. Moreover, after later times post-injection (> 48h), VHH188 appears to display a very low reactivity to C57BL/6 (moTfR+) cells (see Figure 3B). We attribute this to the respective affinities and specificities of both VHHs. We have not investigated the VHH binding kinetics of the mouse versus humanectodomain TfR proteins in vitro. Internalization should be mildly different at best, as ^89^Zr release from DFO occurs with both VHHs in both C57BL/6 and huTfR +/+ mouse models (when injected in a matched configuration). The huTfR +/+ mice rely exclusively on the huTfr for their iron supply. They are healthy with no obvious pathological features. The behavior of the huTfr is therefore presumably similar, if not identical to that of the mouse Tfr, bearing in mind that the huTfr and the mouse Tfr are both reliant on mouse Tf as their ligand

The anti-TfR VHHs were initially developed as a carrier for BBB-transport of VHH-based drug conjugates (previous publications). The data shown here reduces enthusiasm towards this application. Uptake in the brain is several log-factors lower than physiological uptake elsewhere. Potential consequences of off-brain uptake on potential toxicity of VHH-based drug-conjugates could be better emphasized in the discussion.

We did not observe a significant presence of the anti-TfR VHHs in the CNS by PET/CT. We have addressed several possibilities: longer circulation times post-injection may favor transcytosis of the VHHs through the BBB. However, because transcytosis requires endocytosis –^89^Zr may be released by their chelating moiety at this step. The only radiotracers with a covalent bond between the radio-isotope and the VHHs in our work are the ^18^F VHHs, but the signal acquisition window may have been too short to observe transcytosis and accumulation in the CNS. Another possible caveat is that PEGylation of the radiotracers may be an obstacle to transcytosis. The circulatory halflife of unpegylated VHHs is too low to allow adequate visualization after 24 hours postinjection, as the conjugates rapidly clear from the circulation (t ½ = 30 minutes or less). We have updated the discussion to address these points.

In several locations (I have counted 5) a space is missing between words, please double-check.

We carefully checked the manuscript to remove any remaining typos.

It is unclear to me why for the melanoma-tracking experiment the tracer is switched from the 89Zr-labeled variant to the 64Cu-labeled variant.

The decision to switch to the ^64^Cu labeled VHHs for the melanoma experiment stemmed from a wish to (1) evaluate the performance of the ^64^Cu-radioconjugates in detecting transplanted cells as we had done with the ^89^Zr conjugates and (2) assess how the (non-specific) liver signal seen with ^64^Cu contrasts with a specific signal.

typo in discussion: C57BL/6 instead of C57B/6

We have corrected the typo.

It is unclear to me why in FIG1B cells are labeled with 35S. Is it correct that the signals seen are due to staining membranes with anti-TfR mAbs? Or is this an autoradiography of the gel?

In Figure 1B cells were labeled with 35S-Met/Cys, while the images shown are indeed those of Western Blots, using an anti-TfR monoclonal antibody as the primary antibody to detect human and mouse TfR retrieved by the anti Tfr VHHs. Autoradiography using the same lysates showed the presence of contaminants in the VHH eluates, as commonly seen in immunoprecipitates from metabolically labeled cells (as distinct from IP/Westerns). For this reason, we performed a Western Blot on the same samples to confirm TfR pull-down. As written in the results section, we also performed LCMS analysis of the immunoprecipitated proteins to better characterize contaminating proteins (Table 1). To clarify this, we have now added the autoradiographs in supplementary data (supplementary figure 15) and added a reference to these observation in the results.

ROI quantifications in all figures: these should be expressed as %ID/cc instead of %ID/g. Ex vivo tissue counts should be in %ID/g instead of cpm.

We have converted all ROI quantification figures as %ID/cc based on the assumption that 1mL (1cc) = 1g. For ex vivo tissue counts, %ID/g has been calculated based on injected dose (except for figure 3G, where the comparisons in %ID/G are not possible due to the uncertain nature of bone marrow and whole blood). All figures have now been updated.

Fig4: it would be good to also see respective mouse controls (C57BL6 vs hTfR+/+) for the 64Cu- and 18F-labeled VHH123 tracers. Each radiolabeling methodology changes in vivo biodistribution and specificity, which can be better assessed by using appropriate controls.

We had performed these controls but they were not included in the manuscript as deemed redundant with the results of Figure 3. We have now separated Figure 4 in two panels (Figure 4A and 4B) with figure 4A showing the 1h timepoint post-injection of VHH123 radiotracers in C57BL/6 vs huTfr^+/+^ and Figure 4B showing the 24h timepoint in the same configuration. ROI analyses were also done on the huTfR^+/+^ controls and were included in Figure 4C as well.

Fig7: is it correct that mouse imaging is performed at 24h p.i. and dissected embryo's at 72h p.i.? Why are there 2 days between each procedure of the same animals?

We acquired images at di erent timepoints, specifically at 1h, 24h, 48h and 72 hours after radio-tracer injection. As 72 h was the last timepoint, the mice were sacrificed the same day and embryo dissection performed thereafter, at 72 hours post radiotracer injection. We decided to show the 24h timepoint images as they were the most representative of the series, o ering the best signal-to-noise ratio. The signal pattern did not change over the course from 24h to 72h. We have now added those timepoints in the supplementary data.